# I-DRUID: LAYOUT TO IMAGE GENERATION VIA INSTANCE-DISENTANGLED REPRESENTATION AND UNPAIRED DATA

**Fengxiang Yang, Tianyi Zheng, Bangjie Yin, Shice Liu, Peng-Tao Jiang, Jinwei Chen, Bo Li***
vivo BlueImage Lab, vivo Mobile Communication Co., Ltd.
{yangfengxiang, zhengtianyi, libra}@vivo.com

## ABSTRACT

Layout-to-Image (L2I) generation, aiming at coherently generating multiple instances conditioned on the given layouts and instance captions, has raised substantial attention in the recent research. The primary challenges of L2I stem from 1) attribute leakage due to the entangled instance features within attention and 2) limited generalization to novel scenes caused by insufficient image-text paired data. To address these issues, we propose I-DRUID, a novel framework that leverages **i**nstance-**d**isentanglement **r**epresentations (IDR) and **u**npa**i**red **d**ata (UID) to improve L2I generation. IDR are extracted with our instance disentanglement modules, which utilizes information among instances to obtain semantic-related features while suppressing spurious parts. To facilitate disentangling, we require semantic-related features to trigger more accurate attention maps than spurious ones, formulating the instance-disentangled constraint to avoid attribute leakage. Moreover, to improve L2I generalization, we adapt L2I with unpaired, prompt-only data (UID) to novel scenes via reinforcement learning. Specifically, we enforce L2I model to learn from unpaired, prompt-only data by encouraging / rejecting the rational / implausible generation trajectories based on AI feedback, avoiding the need for paired data collection. Finally, our empirical observations show that IDM and RL cooperate synergistically to further enhance L2I accuracies. Extensive experiments demonstrate the efficacy of our method.

## 1 INTRODUCTION

Recent advances in text-to-image (T2I) generation have achieved remarkable success, primarily driven by diffusion models (Rombach et al., 2022; Ho et al., 2020; Esser et al., 2024; Ramesh et al., 2021). By employing UNet (Ronneberger et al., 2015) or multi-modal diffusion transformers (MM-DiT) (Li et al., 2024; Esser et al., 2024) for noise / velocity prediction, these models learn to generate high-quality images with given prompts. To achieve finer control over the generation process, recent works have explored various spatial controls, such as semantic masks (Couairon et al., 2023; Kim et al., 2023; Zhang et al., 2025a), sketches (Voynov et al., 2023; Zhang et al., 2023), or bounding boxes (Zhang et al., 2025b; Zhou et al., 2024a; Wang et al., 2024). Among these methods, bounding box-based control has become a particularly prevalent controlling factor (Li et al., 2023; Wang et al., 2024; Xie et al., 2023; Zhang et al., 2025b; Zhou et al., 2024b) due to its compatibility with downstream vision tasks. This has spurred the development of layout-to-image (L2I) generation (Zhou et al., 2024a; Wang et al., 2024; Xie et al., 2023; Li et al., 2023; Zhang et al., 2025b), which aims to synthesize multiple objects coherently based on a given spatial layout and corresponding captions.

Concurrent studies show promising results in L2I (Zhou et al., 2024a; Zhang et al., 2025b; Wang et al., 2024), but two key challenges remain. (1) The information fusion within attention layer hinders instance rendering (Dahary et al., 2024; Wang et al., 2024), leading to the persistence of attribute leakage. Previous works attempt to address attribute leakage through attention map manipulation (Dahary et al., 2024; Zhou et al., 2024a; Wang et al., 2024), but the inherent difficulty of CLIP (Radford

---

*Corresponding author.

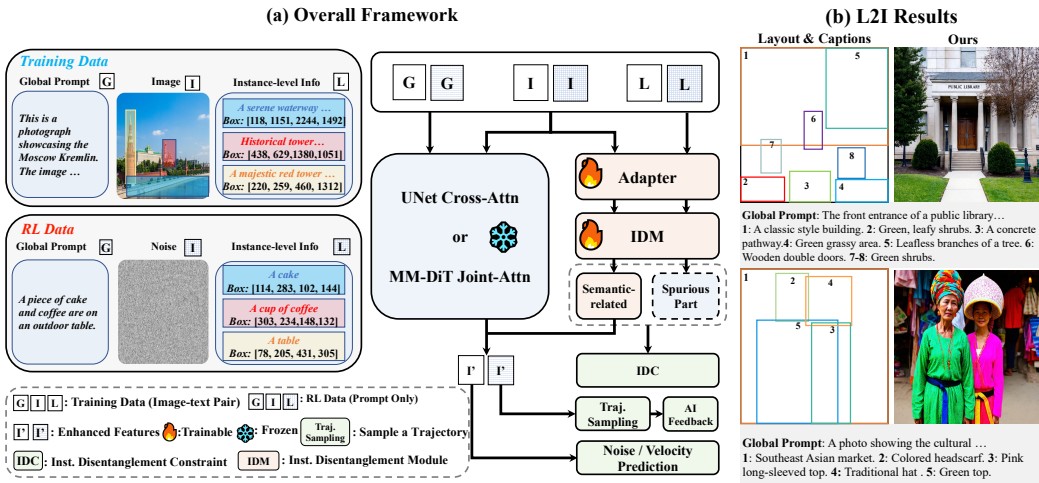

Figure 1: (a) Overall framework of I-DRUID. I-DRUID jointly considers attribute leakage and improves model generalization via AI feedback. (b) Visual results from our method.

et al., 2021) in differentiating singular attributes from complex prompts remains a bottleneck (Feng et al., 2023). Moreover, few studies have considered L2I under MM-DiT scenario (Zhang et al., 2025b). Therefore, it is necessary to explore precise instance-level representation control for both UNet- and MM-DiT-based architectures. (2) L2I models may encounter performance degradation when being deployed to novel scenes. *e.g.*, models trained on training set with long captions (Wang et al., 2024; Zhang et al., 2025b) get poor performance when deployed under testing set with short coarse captions (Zhou et al., 2024b), demonstrating poor generalization when training with limited pair-wise training data. Collecting more pair-wise data is an empirical solution, but suffers from huge time cost. Inspired by the success of RL in scaling large language models to complex tasks (Guo et al., 2025; Jaech et al., 2024), we would like to advance L2I with RL by using prompt-only unpaired data, letting the novel-scene adaptation to be exempted from pair-wise data collection.

As shown in Fig. 1-(a), we present I-DRUID, which is comprised of instance disentanglement learning, coupled with reinforcement learning. (1) We first introduce adapters (Ye et al., 2023) to inject layout information into training process, then decompose attention features into "semantic-related features" and "spurious part" with our instance disentanglement modules (IDM). The core of disentanglement through IDM is based on our instance disentangling constraint (IDC). Our key insight of designing IDC is that semantic-related features must elicit more precise attention maps than spurious features, thereby facilitating the disentanglement process during L2I optimization. (2) To generalize our L2I model to OOD prompts, we further introduce a novel reinforcement learning (RL) framework (Agarwal et al., 2019; Liu et al., 2025b), encouraging L2I model to learn from unpaired novel prompts based on AI feedback. Specifically, we conduct trajectory sampling from Gaussian noise and the given novel prompt, producing images for the assessment of visual language model (VLM) in terms of spatial accuracy and instance consistency in an online manner. The feedback is used as a guidance in analyzing best generation policy, thus encouraging L2I capability. (3) We also demonstrate IDC and RL could mutually benefit each other as IDM provides a more accurate generation policy for RL, advancing L2I results as shown in Fig. 1-(b). Our method also has a high flexibility, which could be easily utilized for both UNet-based (*e.g.*, SD 1.5) and MM-DiT-based architectures (*e.g.*, SD3).

To sum up, our contributions are three-fold:

- We seek instance-disentangled representation, which is achieved with our IDC-supervised IDM. IDC leverages collective instance information to extract semantic-related features, which triggers more precise attention maps and thus avoids attribute leakage.

- We equip our L2I model with a novel Reinforcement Learning (RL) strategy to improve its generalization. RL enables L2I to learn from unpaired data, letting novel-scene adaptation to be exempted from pair-wise data collection.

- We demonstrate that these two components can be synergistically cooperated to further enhance L2I accuracy under both UNet and MM-DiT-based architectures. Our approach achieves state-of-the-art results on multiple L2I benchmarks, demonstrating its efficacy and flexibility.

## 2 RELATED WORK

**Layout to Image Generation**. Diffusion models like SD (Ho et al., 2020; Rombach et al., 2022), SD3 (Esser et al., 2024), and FLUX (Labs, 2024) are powerful tools to achieve text to image generation. To achieve fine-level control, additional spatial control like bounding boxes (Wang et al., 2024; Li et al., 2023; Zhou et al., 2024b;a; Zhang et al., 2025b; Lee et al., 2024) are introduced into generation process to craft entities within the given location and instance-level prompts, termed as layout to image generation. Generally speaking, these methods can be categorized as training-based (Zhou et al., 2024b; Li et al., 2023) and training-free approach (Xie et al., 2023; Lee et al., 2024). The former introduce adapters (Ye et al., 2023; Mou et al., 2023) and additional modules to encode locations for subsequent attention. The latter focus on manipulating attention map at test-time to achieve location control for each entity. Although effective, most of them are conducted based on UNet architecture, while ignoring modern MM-DiT architectures like SD3 (Esser et al., 2024). Recently, Creati-Layout (Zhang et al., 2025b) first devise SD3-based L2I scheme, but ignores explicit constraint to alleviate attribute leakage (Wang et al., 2024). Different from previous methods, our approach devise disentanglement modules to advance L2I task under both UNet-based and MM-DiT-based scenarios.

**Reinforcement Learning**. Reinforcement learning (RL) (Schulman et al., 2017) is a widely-used strategy to align model's response to human preference, both for large language models (Rafailov et al., 2023; Wang et al., 2025; Yan et al., 2024) and diffusion models (Chen et al., 2024; Fan et al., 2023). The core idea of RL is training a model to interact with environment, typically a reward model, that provides feedback for the model's responses. This feedback guides the model in exploring and learning an optimal policy. Early RL methods for alignment often relied on either a separate reward model (Ramamurthy et al., 2022; Peng et al., 2019) or extensive human annotation of samples (Rafailov et al., 2023; Yuan et al., 2024) to obtain preference data. However, the recent development of large-scale models has led to a paradigm shift. Learning from AI feedback has emerged as a promising and efficient alternative for scaling the alignment process (Bai et al., 2022; Fan et al., 2023). The RL under diffusion scenario can be categorized into online and offline scheme. DPOK (Fan et al., 2023) and DDPO (Black et al., 2023) are pioneering works that introduce RL into image generation by fine-tuning generation policy with feedback from AI like ImageReward (Xu et al., 2024) in an online manner. DiffDPO (Wallace et al., 2024) proposes the first DPO-based (Rafailov et al., 2023) RL method to fine-tune diffusion in an offline manner. Although these methods show promising results, most of them relies on the randomness of generation trajectory to achieve environmental exploration, which is not compatible with SD3 (Esser et al., 2024) or FLUX (Labs, 2024) with deterministic ODE forward process. Recently, the advent of SDE-based forwarding equivalence (Albergo et al., 2023; Liu et al., 2025a) enables efficient exploration during the forwarding process of flow-matching methods. However, the application of RL under L2I scenario is still under-explored, inspiring us to scale L2I model with the help of AI feedback and RL.

## 3 METHODOLOGY

### 3.1 OVERVIEW

**Problem Formulation**. Layout to image generation (L2I) assumes the users to give the following information during inference. (1) a global prompt $p_G$ defining the overall semantic information for the generated image; (2) $n$ instance captions $\mathcal{P}_L = \{p_1, p_2, ..., p_n\}$, which describes detailed information for each instance; (3) layout $\mathcal{B} = \{b_1, b_2, ..., b_n\}$, where $b_i = \{x_{i0}, y_{i0}, x_{i1}, y_{i1}\}, (1 \leq i \leq n)$ containing top-left coordinate $(x_{i0}, y_{i0})$ and bottom-right coordinate $(x_{i1}, y_{i1})$ for each instance. The goal of L2I is generating images that follows the above instructions. Although our method is capable of L2I for both UNet-based (*e.g.*, SD-1.5) and MM-DiT-based architectures (*e.g.*, SD3), *we mainly introduce our method with SD3 and the SD-1.5 variant could be easily derived.*

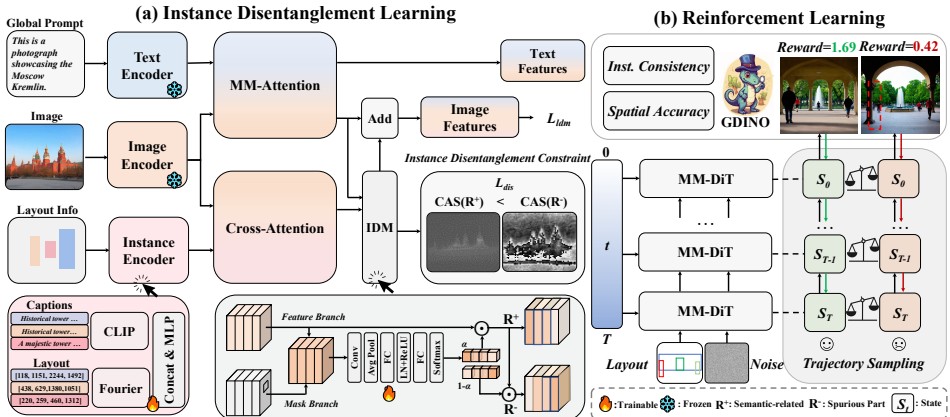

Figure 2: Overall Training Process. Our method is comprised of: (a) Instance Disentanglement Learning and (b) Reinforcement Learning Stage. At (a), our instance disentanglement module (IDM) is supervised by instance disentanglement constraint (IDC) for extracting semantic-related representations $\mathbf{R}^+$, while discarding spurious part $\mathbf{R}^-$ based on the proposed metric CAS. Moreover, we sample various generation trajectories with unpaired data and adopt GDINO to evaluate generated images. Better results will be encouraged at (b), adapting L2I model to novel scenes.

**Overall Pipeline**. As shown in Fig. 2-(a), Our L2I solution is comprised of (a) instance disentanglement learning and (b) reinforcement learning. The former is built upon adapter-based L2I (Li et al., 2023; Wang et al., 2024), further refining features based on our IDM and IDC. The latter enforces L2I model to seek for the optimal generation policy based on the given prompt and feedback from reward, thus generalizing the model to novel scenes without using image-text paired data. These two contributions mutual benefit each other and further improves L2I capability. Next, we introduce our solution in detail.

## 3.2 INSTANCE DISENTANGLEMENT STAGE

**Instance Disentanglement Module**. Previous L2I methods (Li et al., 2023; Wang et al., 2024; Zhang et al., 2025b) achieve layout control by injecting layout-caption coupled features with cross-attention for optimization. However, as demonstrated in previous literature (Dahary et al., 2024; Zhou et al., 2024b;a), attention tends to fuse instance-level features, leading to the attribute leakage. We therefore devise IDM as feature refiner to alleviate this. The IDM specifications are shown in Fig. 2-(a), which receives $n + 1$ enhanced features and layout masks as inputs. IDM learns to extract "semantic-related features" $\mathbf{R}^+$ while discarding "spurious part" $\mathbf{R}^-$ for subsequent generation. Specifically, we compute the given $n$ instances' enhanced features with corresponding layout and captions, coupled with global-prompt-enhanced features to formulate $n + 1$ inputs $\mathcal{E} = \{\mathbf{e}_1, \mathbf{e}_2, ..., \mathbf{e}_{n+1}\} \in \mathbb{R}^{(n+1) \times C \times W \times H}$. These features formulate the input of *feature branch*. *For mask branch*, we obtain layout masks for instance $i$ with its bounding box $b_i = \{x_{i0}, y_{i0}, x_{i1}, y_{i1}\}$. Pixels within bounding box $b_i$ will be assigned with 1 while others are set to 0. The mask for global prompt is set to all-one matrix, formulating $n + 1$ masks $\mathcal{M} = \{\mathbf{m}_1, \mathbf{m}_2, ..., \mathbf{m}_{n+1}\}$. IDM jointly considers $\mathcal{E}$ and $\mathcal{M}$ to get the optimal channel-wise weighting scheme:

$$\alpha = \text{IDM}(\mathcal{E}, \mathcal{M}) \in \mathbb{R}^{n+1}, \tag{1}$$

where $\alpha$ is the weighting scores assigned to each enhanced feature in $\mathcal{E}$. We thus obtain "semantic-related features" $\mathbf{R}^+$ while discarding "spurious part" $\mathbf{R}^-$ through simple multiplication:

$$\mathbf{R}^+ = \alpha \odot \mathcal{E}; \quad \mathbf{R}^- = (1 - \alpha) \odot \mathcal{E}, \tag{2}$$

where $\mathbf{R}^+$ and $\mathbf{R}^-$ are "semantic-related features" and "spurious part" with the size of $(n + 1) \times C \times W \times H$.

**Instance Disentanglement Constraint**. To facilitate the disentangling process, we further devise instance disentanglement constraint (IDC) by measuring $\mathbf{R}^+$'s ability in triggering accurate attention map. Specifically, $\mathbf{R}^+$ will be used to obtain cross-attention map $\mathbf{R}_{CA}^+$. We then adopt layout mask $\mathcal{M}$ to extract attention value at background and evaluate overall attention accuracy with the following

criterion:

$$\mathrm{CAS}(\mathbf{R}_{CA}^+, \mathcal{M}) = \sum_{i=1}^{n} |\mathbf{R}_{CA,i}^+ - \mathrm{AVG}(\mathbf{R}_{CA,i}^+ \odot (1 - \mathcal{M}_i))| \odot (1 - \mathcal{M}_i), \quad (3)$$

where $\mathcal{M}_i$ and $\mathbf{R}_{CA,i}^+$ are the mask and triggered attention map for instance $i$, respectively. $\mathrm{AVG}(\cdot)$ is the averaging operation. In our design, $\mathrm{AVG}(\mathbf{R}_{CA,i}^+ \odot (1 - \mathcal{M}_i))$ extracts instance $i$'s averaged background attention values. Eq. 3 subsequently quantifies the total absolute deviation beyond instance $i$'s scope. Higher CAS indicates potential bounding box mis-alignment for the generated instance as the instance will lead to high dispersion at background. Based on CAS, it is obvious that $\mathbf{R}^+$ should trigger lower CAS value than $\mathbf{R}^-$, formulating an inequality $\mathrm{CAS}_{\mathbf{R}^+} < \mathrm{CAS}_{\mathbf{R}^-}$. The inequality could be further transformed into our "Instance Disentanglement Constraint":

$$L_{dis}(\mathbf{R}_{CA}^+, \mathbf{R}_{CA}^-, \mathcal{M}) = \mathrm{Softplus}\Big[\mathrm{CAS}(\mathbf{R}_{CA}^+, \mathcal{M}) - \mathrm{CAS}(\mathbf{R}_{CA}^-, \mathcal{M})\Big], \quad (4)$$

where $\mathrm{Softplus} = \ln(1 + \exp(\cdot))$ is a monotonically increasing function. Minimizing Eq. 4 is equivalent to encourage lower CAS score for $\mathbf{R}^+$ and higher CAS for $\mathbf{R}^-$, thus enabling our disentanglement to heuristically learn semantic-related features for L2I generation.

## 3.3 REINFORCEMENT LEARNING STAGE

**RL Formulation**. We build our algorithm based on classical PPO (Schulman et al., 2017). The elements of conventional RL could be summarized into four essentials: (1) state (2) action (3) policy (4) reward. We then introduce these elements in the context of flow matching model SD3. For the given time $t$ and condition $\mathbf{y} = \{\mathbf{b}, \mathbf{p}\}$ ($\mathbf{b}$ and $\mathbf{p}$ are bounding box and their corresponding instance captions), "state" is $\mathbf{s}_t \triangleq (\mathbf{x}_t, t, \mathbf{y})$ "action" denotes denoised latent at $t-1$, i.e., $\mathbf{a}_t \triangleq \mathbf{x}_{t-1}$. "Policy" denotes the transition probability between two time steps $\pi(\mathbf{a}_t|\mathbf{s}_t) \triangleq \pi(\mathbf{x}_{t-1}|\mathbf{x}_t, \mathbf{y})$. By inferring from $T$ to $0$, the model generate final output $\mathbf{x}_0$, which could be subsequently decoded by VAE to obtain image $\mathbf{X}_0$, and evaluated with reward model $r(\cdot)$ in terms of spatial accuracies and instance consistency. $r(\cdot)$ assigns score as a guidance to supervise the overall generation process, thus promoting the L2I accuracy when novel prompts emerges. For UNet-based diffusion model, policy is derived based on DDPM (Ho et al., 2020), which contains randomness to achieve exploration. However, MM-DiT-based diffusion is built upon flow matching (Esser et al., 2024), which relies on deterministic ODE and hinders environmental exploration. We thus divert deterministic ODE sampling to SDE to formulate our policy. Specifically, we follow (Domingo-Enrich et al., 2024; Albergo et al., 2023; Liu et al., 2025a) and change the policy as follow:

$$\mathbf{x}_{t+\Delta t} = \mathbf{x}_t + \Big[v_\theta(\mathbf{x}_t, t, \mathbf{y}) + \frac{\sigma_t^2}{2t}(\mathbf{x}_t + (1-t)v_\theta(\mathbf{x}_t, t, \mathbf{y}))\Big]\Delta t + \sigma_t\sqrt{\Delta t}\epsilon, \quad (5)$$

where $\mathbf{x}_t$ is the partially denoised image latent at time $t$, $v_\theta$ is the predicted velocity. $\epsilon$ is the random term sampled from standard Gaussian distribution $\mathcal{N}(0, 1)$. $\sigma_t = a\sqrt{\frac{t}{1-t}}$ and $a = 0.7$ is a hyper-parameter defined in (Liu et al., 2025a).

**Reward Definition**. Considering the layout-control nature of L2I task, we choose Grounding-DINO (GDINO) (Liu et al., 2024) as the reward model $r(\cdot)$ to assess the overall spatial accuracy and instance consistency for generated images. GDINO receives instance captions and $\mathbf{X}_0$ to detect the location of objects within the input, then returns detected bounding box and confidence score $\mathbf{o}_{pred} = \{\mathbf{b}_{pred}, \mathbf{c}_{pred}\}$. We thus define the reward based on $\mathbf{o}_{pred}$ and ground-truth $\mathbf{o}$:

$$r(\mathbf{o}, \mathbf{o}_{pred}) = \sum_i \Big[\mathrm{IoU}(b_{pred,i}, b_i) + c_{pred,i}\Big], \quad (6)$$

where $b_{pred,i}$ is the ground-truth bounding box for instance $i$, $c_{pred,i}$ is the confidence score in detecting instance $i$ based on instance caption $p_{pred,i}$. IoU is the function to compute IoU score between ground-truth bounding box and GDINO-predicted counterpart. Based on these formulations, we require L2I model to perform actions and interact with environment, seeking the optimal action by optimizing underlying policy. Following standard RL formulation, we term our diffusion model as "actor net".

**Critic-Net**. During RL fine-tuning, we also introduce critic-net $\phi$ as a collaborator to further improve actor-net's generation capability. The critic-net is a light-weighted MLP, which receives model state $s_t$ and embedded user inputs $\mathbf{y}$ to predict final scalar reward value. Its training goal is minimizing the discrepancy between predicted and actual reward given by GDINO to learn the average reward:

$$L_{critic}(s_t, \mathbf{o}, \mathbf{o}_{pred}) = \Big[ \phi(s_t) - r(\mathbf{o}, \mathbf{o}_{pred}) \Big]^2. \tag{7}$$

With critic-net, we could further define advantage score by subtracting critic-net's prediction from reward:

$$A(s_t, \mathbf{o}, \mathbf{o}_{pred}) = r(\mathbf{o}, \mathbf{o}_{pred}) - \phi(s_t), \tag{8}$$

the advantage function measures the expected reward over the average performance, which facilitates actor-critic collaboration.

**Actor-Critic Collaboration via PPO**. The goal of RL is fine-tuning actor-net to obtain high advantage scores for generated images. By following classical PPO (Schulman et al., 2017) algorithm, we use importance sampling to optimize actor net. Specifically, we compute probability ratio $\rho_t$ at $t$-th step by comparing the policy for current and old model:

$$\rho_t = \frac{\pi(a_t | s_t, \mathbf{y})}{\pi(a_t^{old} | s_t^{old}, \mathbf{y})}, \tag{9}$$

where $a_t^{old}$ and $s_t^{old}$ are action and state for old actor net, which is commonly the EMA model (Klinker, 2011). The final loss for actor-net could be formulated as:

$$L_{rl} = \max \Big[ -\rho_t A_t, -\text{clip}(\rho_t, 1-\zeta, 1+\zeta)A_t \Big] + \text{KL}(\pi(a_t | s_t, \mathbf{y}) || \pi(a_t^{old} | s_t^{old}, \mathbf{y})), \tag{10}$$

where $\text{clip}(\cdot)$ is the clipping function, limiting the probability ratio in the range of $[1\text{-}\zeta, 1\text{+}\zeta]$ to ensure stable training (Schulman et al., 2017), $\text{KL}(\cdot||\cdot)$ is the KL divergence to avoid training collapse. It should be noted that $L_{rl}$ instructs and optimizes actor-net through $\rho_t$ to achieve the enhancement of advantage scores $A_t$. $A_t$ does not directly supervise actor-net's training.

**Joint Optimization**. We incorporate flow matching loss (Esser et al., 2024) to to optimize our actor-net:

$$L_{ldm} = \mathbb{E}_{\epsilon \sim \mathcal{N}(0,1), t, \mathbf{y}, \mathbf{x}_t} || \mathbf{v} - \mathbf{v}_\theta(\mathbf{x}_t, t, \mathbf{y}) ||_2^2, \tag{11}$$

where $\mathbf{v}$ is the ground-truth velocity, obtained through encoded image and Gaussian noise. For UNet-based diffusion model like SD 1.5 (Rombach et al., 2022), the objective of training should be noise prediction (Ho et al., 2020). We will introduce our UNet-based solution in the supplementary. The final loss for actor-net's optimization are formulated as follow:

$$L_{act} = L_{ldm} + \lambda_{dis}L_{dis} + \lambda_{rl}L_{rl}, \tag{12}$$

where $\lambda_{dis}$ and $\lambda_{rl}$ are balancing factors. In practice, we utilize model pool $\mathcal{MP}$ to collect old model's policies. $\mathcal{MP}$ is a memory pool, storing old model's policies to facilitate RL. When computing Eq. 12, we randomly sample an old policy for image generation and optimization. The overall training pipeline is shown in our appendix, at Alg. 1.

**RL Speed-up**. Our original RL scheme requires trajectory sampling for reward computation and action-based fine-tuning, consuming large amount of time due to the inference of model with full time-steps. Recently, the finding that optimization at early time steps (Zhou et al., 2024c; Kang et al., 2025; Zheng et al., 2025) yields similar or even better results inspiring us to conduct RL at early time steps. We therefore conduct RL only at the first 20% time steps and save corresponding actions to avoid reference model's action computation. Moreover, our PPO-based RL naturally has computational advantage over the popular GRPO-based methods (Liu et al., 2025a) as GRPO requires sampling several trajectories at the same time for group-level advantage computation.

## 4 EXPERIMENT

### 4.1 EXPERIMENTAL SETTING

**Training Sets**. We train our model with LayoutSAM (Zhang et al., 2025b) and COCO-2014-MIG (Zhou et al., 2024b). LayoutSAM contains 2.7 million image-text pairs and 10.7 million

Table 1: Comparison with state-of-the-art methods on COCO-MIG. We compare our method with state-of-the-art generation methods, including BoxDiff (Xie et al., 2023), Reco (Yang et al., 2023), GLIGEN (Li et al., 2023), RichContext Cheng et al. (2024b), InstanceDiff (Wang et al., 2024), MIGC (Zhou et al., 2024b), and Creati-Layout (Zhang et al., 2025b). *: Evaluated based on MIGC's code. Red and Green are best and second best results.

| Methods | Instance Success Rate ↑ | | | | | | mIoU ↑ | | | | | |
|---|---|---|---|---|---|---|---|---|---|---|---|---|
| | L2 | L3 | L4 | L5 | L6 | Avg | L2 | L3 | L4 | L5 | L6 | Avg |
| BoxDiff | 24.61 | 19.22 | 14.20 | 11.92 | 9.31 | 15.85 | 32.54 | 29.88 | 25.39 | 23.81 | 21.19 | 26.56 |
| GLIGEN | 42.30 | 35.55 | 32.66 | 28.18 | 30.84 | 33.89 | 37.58 | 32.34 | 29.95 | 26.60 | 27.70 | 30.83 |
| RichContext | 40.31 | 30.83 | 30.78 | 26.50 | 25.42 | 30.76 | 37.88 | 31.43 | 30.35 | 28.42 | 26.59 | 30.93 |
| InstanceDiff | 58.00 | 52.16 | 55.03 | 47.59 | 47.12 | 51.98 | 52.14 | 48.64 | 50.36 | 42.64 | 42.86 | 47.33 |
| MIGC | 67.70 | 59.61 | 58.09 | 56.16 | 56.88 | 59.68 | 59.39 | 52.73 | 51.45 | 49.52 | 49.89 | 52.60 |
| Reco | 65.50 | 56.10 | 52.30 | 52.40 | 58.30 | 56.90 | 55.70 | 46.70 | 47.20 | 43.30 | 48.80 | 47.60 |
| Creati-Layout* | 65.93 | 65.41 | 56.40 | 50.62 | 50.00 | 57.67 | 56.61 | 56.29 | 50.30 | 45.85 | 45.66 | 50.94 |
| Ours (SD-1.5) | 79.10 | 70.24 | 65.48 | 63.87 | 66.97 | 69.13 | 70.61 | 62.10 | 58.63 | 56.04 | 58.18 | 68.18 |
| Ours (SD-3) | 76.87 | 69.16 | 62.96 | 52.37 | 52.39 | 62.75 | 63.35 | 59.01 | 53.80 | 51.32 | 50.54 | 55.60 |

fine-grained instance-level captions derived from SAM (Kirillov et al., 2023) dataset. Different from LayoutSAM, COCO-2014-MIG (Zhou et al., 2024b) adopts stanza (Qi et al., 2020) to recognize entities, while obtaining instance-level bounding boxes with Grounding-DINO. Therefore COCO-2014-MIG has coarse instance captions while LayoutSAM has detailed insantce captions. We only use the prompt (layout and instance captions) from COCO-2014-MIG for RL while adopt image-text pairs from LayoutSAM for diffusion training.

**Evaluation Protocol**. We evaluate our method on two large-scale benchmarks: COCO-MIG (Zhou et al., 2024b) and LayoutSAM-eval (Zhang et al., 2025b).COCO-MIG consists of 800 multi-instance prompts from COCO-2014, spanning five complexity levels (L2–L6) based on instance counts. Following (Zhou et al., 2024b), we use Grounded-SAM (Ren et al., 2024) to assess spatial correctness via Instance Success Rate (ISR), which measures the percentage of successfully detected instances, and mIoU, which reflects the average maximum IoU across all instances. LayoutSAM-eval comprises 5, 000 prompts for comprehensive L2I evaluation. We adopt the MiniCPM-based (Yao et al., 2024) inquiry protocol to evaluate spatial, color, texture, and shape attributes. Furthermore, we report FID, PickScore (Kirstain et al., 2023), and IS to measure the general image quality.

**Implementation Details**. We validate our method on both SD-1.5 (Rombach et al., 2022) and SD3-mid (Esser et al., 2024). Conducting paired image-text training with LayoutSAM while un-paired RL with COCO-2014-MIG. *The optimized model is evaluated on COCO-MIG to check OOD generalization while assessed on LayoutSAM-eval to check basic L2I capability.* For SD-1.5 variant, please check our supplementary for details. For SD3-based variant, we deploy our IDM and IDC on every joint-attention transformer layers. During training, we set training batch size as 128, RL batch size as 16, and optimize the model for 20 epochs with a learning rate of $1 \times 10^{-4}$, which will take 4 days on 8 NVIDIA H20 GPUs. Note that RL will not be included into training for the first 10 epochs. During RL, the critic-net and actor-net are optimized in a GAN-like manner. Critic-net will be optimized for 5 times before the collaboration with actor-net. During the inference, we set classifier free guidance scale as 7.5. We follow the inference-time trick used in (Zhang et al., 2025b), activating IDM for the first 30% time steps during inference, while conduct vanilla joint attention for the rest 70%. We set $\lambda_{dis} = \lambda_{rl} = 1$ and $\zeta = 1 \times 10^{-4}$. We set training resolution to $1024 \times 1024$ for SD3. We set maximum number of instances as 10 for each training sample, *i.e.*, only the first 10 instances will be kept during training. Samples with less than 10 instances will be padded with zero for both text and layout embeddings.

## 4.2 COMPARISON WITH STATE-OF-THE-ARTS

**Results on COCO-MIG**. We first report the results on COCO-MIG to check L2I generalization. As shown in Tab. 1, we have two conclusions. (1) Previous methods are struggling in adapting to novel L2I scenes. Specifically, InstanceDiff (Wang et al., 2024) and Creati-Layout (Zhang et al., 2025b) are trained with detailed instance captions, which is different from MIGC (Zhou et al., 2024b) that is trained with short captions. As COCO-MIG assigns short captions for each instance, both

Table 2: Comparison on LayoutSAM-eval. We compare our method with InstanceDiff (Wang et al., 2024), Ranni (Feng et al., 2023), BeYourself (Dahary et al., 2024), MIGC (Zhou et al., 2024b), HiCo (Cheng et al., 2024a), and CreatiLayout (Zhang et al., 2025b).

| Methods | Spatial ↑ | Color ↑ | Texture ↑ | Shape ↑ | FID ↓ | PickScore ↑ | IS ↑ |
|---|---|---|---|---|---|---|---|
| Ranni | 41.38 | 24.10 | 25.57 | 23.35 | 27.24 | 20.49 | 19.81 |
| BeYourself | 53.99 | 31.73 | 35.26 | 32.75 | 28.10 | 20.20 | 17.98 |
| MIGC | 85.66 | 66.97 | 71.24 | 69.06 | 21.19 | 20.71 | 19.65 |
| InstanceDiff | 87.99 | 69.16 | 72.78 | 71.08 | 19.67 | 21.01 | 20.02 |
| HiCo | 87.04 | 69.19 | 72.36 | 71.10 | 22.61 | 21.70 | 20.15 |
| Creati-Layout | 92.67 | 74.45 | 77.21 | 75.93 | 19.10 | 22.02 | 22.04 |
| Ours (SD-1.5) | 86.95 | 70.49 | 73.56 | 72.30 | 22.92 | 21.98 | 17.95 |
| Ours (SD-3) | **93.14** | **75.37** | **78.35** | **77.20** | **17.21** | **23.16** | **22.45** |

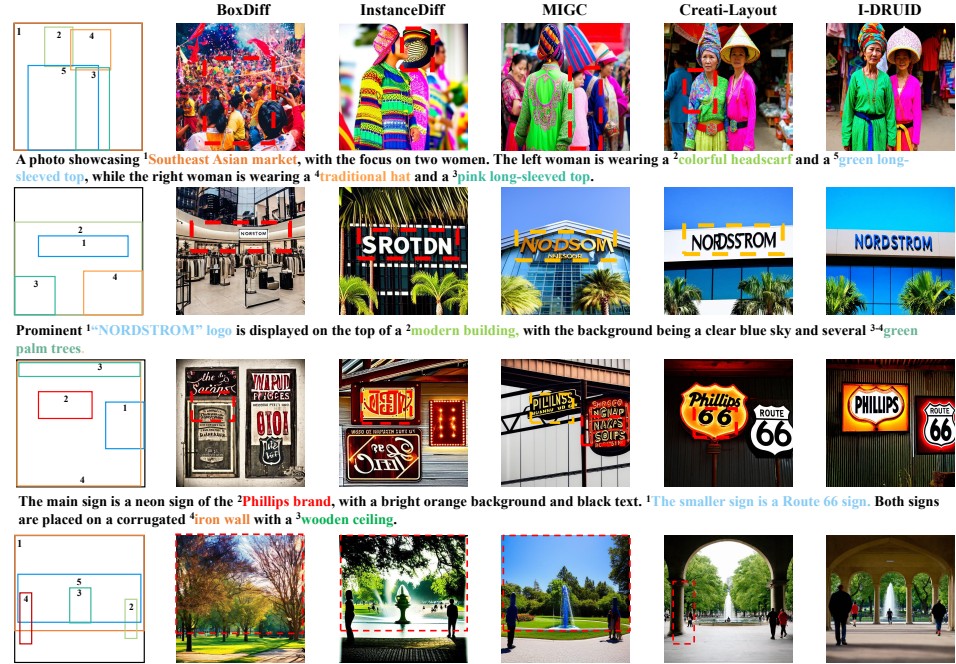

Figure 3: Visualization of generated samples with I-DRUID and state-of-the-arts. Dashed red box: wrongly generated. Dashed yellow box: Distorted text.

InstanceDiff and Creati-Layout fail to outperform MIGC, demonstrating the necessity of improving L2I model's generalization. (2) Our method outperforms all previous works in terms of averaged ISR and mIoU under both SD-1.5 and SD3 scenario, showing its strong flexibility and L2I generalization. By introducing unpaired data from COCO-2014-MIG, I-DRUID learns to generate with short captions under the guidance from GDINO and thus enhances its L2I generalization in novel COCO-MIG.

**Results on LayoutSAM-eval**. We also evaluate our method on LayoutSAM-eval in terms of spatial, color, texture, and shape to show its basic L2I capability. As LayoutSAM-eval provides image-text pair, we also report the general image quality metrics like FID at Tab. 2. It is obvious our SD3 counterpart outperforms other methods. SD3 provides better image quality, enabling our method to achieve lower FID scores on LayoutSAM-eval. Moreover, our method also outperforms other state-of-the-arts in all aspects (spatial, color, texture, and shape), demonstrating its efficacy on large-scale L2I benchmark. It should be noted that MIGC (Zhou et al., 2024b) fails to achieve better results than InstanceDiff (Wang et al., 2024) and Creati-Layout (Zhang et al., 2025b), demonstrating the significance of improving L2I's generalization to different scenes, especially when there is large domain gap between training and testing sets.

Table 3: Ablation study. We report the averaged ISR on COCO-MIG and four main criteria on LayoutSAM-eval to check each component's efficacy. Green and red are best and second best results.

| No. | Attributes | | | | COCO-MIG | LayoutSAM-eval | | | |
|---|---|---|---|---|---|---|---|---|---|
| | IDM | RL-PPO | RL-GRPO | SFT | Avg ISR | Spatial | Color | Texture | Shape |
| 1 | ✗ | ✗ | ✗ | ✗ | 56.82 | 86.96 | 71.08 | 73.20 | 72.44 |
| 2 | ✓ | ✗ | ✗ | ✗ | 57.64 | 88.53 | 73.12 | 74.93 | 75.62 |
| 3 | ✓ | ✗ | ✗ | ✓ | 66.92 | 89.75 | 72.21 | 74.33 | 73.19 |
| 4 | ✓ | ✗ | ✓ | ✗ | 61.64 | 92.86 | 74.67 | 78.65 | 75.80 |
| 5 | ✗ | ✓ | ✗ | ✗ | 60.23 | 91.47 | 72.45 | 74.79 | 72.93 |
| 6 | ✓ | ✓ | ✗ | ✗ | 62.75 | 93.14 | 75.37 | 78.35 | 77.20 |

## 4.3 VISUAL COMPARISON

We visualize some L2I results and compare them with state-of-the-arts for an intuitive evaluation. As shown in Fig. 3, we conclude that (1) Our method achieves better image quality for L2I. Specifically, our method and SD3-based Creati-Layout generate human face and characters with high quality, while other methods like MIGC fail to achieve this. (2) Our method achieves better detail interpretation. *e.g.*, at the third row of Fig. 3, I-DRUID correctly rendered the given instance prompt "Phillips brand", while Creati-layout additionally generates "66", which is potentially caused by the attribute leakage problem. By explicitly considering the instance-disentangled representation for L2I, the problem is alleviated. Therefore, we demonstrate that our method show competitive capability in L2I.

## 4.4 ABLATION STUDY

**Ablated Visual Comparison**. We gradually add each component of our proposed method and check the direct influence of each component on the final generated images through visualization. The results are shown in Fig. 4. From the visualization, attribute leakage could be easily found when solely using $L_{ldm}$ for optimization (at the column of "$L_{ldm}$"). Specifically, at the first row of Fig. 4, "a brown hot dog" is mistakenly generated as a red hot dog due to nearby instance prompt "a red hot dog". Similar results could also be noted at the second row, where the "white chair" is mistakenly generated as "a brown chair". After introducing $L_{dis}$, the attribute leakage problem is alleviated as all instances are generated in their corre-

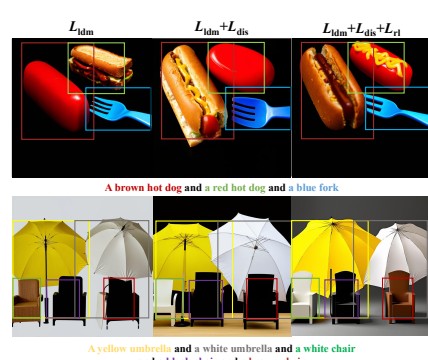

Figure 4: Ablated Visual Comparison.

sponding locations. Moreover, after adding $L_{rl}$, the L2I model achieves fine-level attribute control for instances. *e.g.*, at the second row, the instance "a brown chair" adjust its color to align with the attribute "brown".

**Effectiveness of using IDM**. To check the efficacy of each component, we also conduct ablation study on COCO-MIG and LayoutSAM-eval and report the results in Tab. 3. We mainly check four attributes of our method. "IDM": using our IDM+IDC to disentangle features during L2I; "RL-PPO": PPO fine-tuning; "RL-GRPO": replacing PPO with GRPO. GRPO does not require critic net for advantage estimation, but needs to sample multiple trajectories at the same time; "SFT": We annotate COCO-2014-MIG with stanza (Qi et al., 2020) and GDINO in an offline manner while integrating the annotated data into training. *Note that SFT introduces images from COCO-2014-MIG for joint optimization and thus cannot be conducted with unpaired data.* By comparing "No. 1" and "No. 2", we found that introducing IDC is beneficial to improving L2I accuracies on both benchmarks, yielding improvement on L2I accuracies on all metrics. The results indicate IDC's effectiveness in generating multiple samples with accurate attribute and layout.

**Effectiveness of RL**. By further comparing the results between "No. 1", "No. 3", "No. 4", "No. 5" and "No. 6", we conclude RL is beneficial to improving L2I accuracies. Specifically, we only adopt unpaired prompt-only data from COCO-2014-MIG in our RL-based experiments, but the results on COCO-MIG demonstrate RL's efficacy in improving L2I model's generalization with unpaired data. It should also be noted that "SFT" variant in "No. 3" is optimized with paired data

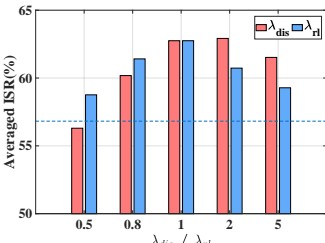
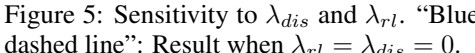
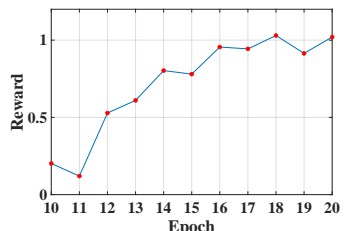

Figure 5: Sensitivity to $\lambda_{dis}$ and $\lambda_{rl}$. "Blue dashed line": Result when $\lambda_{rl} = \lambda_{dis} = 0$.

Figure 6: Reward stability.

from COCO-2014-MIG and violates our experimental setting that requires unpaired data for training. Although "SFT" achieves better results on COCO-MIG, it fails to enhance L2I accuracies under LayoutSAM-eval. Therefore, *SFT may lead to performance degradation due to the domain gap between different training sets*. Moreover, introducing RL does not lead to performance degradation, which further demonstrates RL's efficacy.

**Effectiveness of using Different RL scheme**. We also compare PPO-based RL with GRPO-based (Shao et al., 2024; Liu et al., 2025a) RL. By comparing "No. 4" and "No. 6", we note that using different RL strategies yields similar improvement over baseline. The results demonstrate the efficacy of RL even when different RL strategies. Moreover, GRPO-based counterpart requires sampling several trajectories at the same time to obtain group-level advantage, significantly increasing the training time cost. As advantage in our solution is obtained through light-weighted critic-net, training under our scheme is much faster than GRPO-based method. Our method achieves training speed of nearly 15s / iter on NVIDIA H20, while GRPO-based method needs more than 60s / iter when setting group size to 4 and conducting RL on full time steps.

## 4.5 SENSITIVITY ANALYSIS

**Sensitivity to $\lambda_{dis}$ and $\lambda_{rl}$.** We first check model's sensitivity to hyper-parameters $\lambda_{dis}$ and $\lambda_{rl}$ with averaged ISR over all instance-levels. At this experiment, we change $\lambda_{dis}$ / $\lambda_{rl}$ from 0.5 to 5 while keeping $\lambda_{rl}$ / $\lambda_{dis}$ at 1. The results are shown in Fig. 5, we visualize baseline method's averaged ISR (results of "No.1" in Tab. 3) with blue dashed line. From the results, we draw two conclusions. (1) Our method is not sensitive to both hyper-parameters. Specifically, we note in Fig. 5 that changing these two parameters does not bring significant degradation on performance and the best results are achieved when setting $\lambda_{dis} = \lambda_{rl} = 1$. (2) $\lambda_{dis}$ plays an important role in our L2I algorithm. Specifically, we note performance degradation than baseline when setting $\lambda_{dis} = 0.5$ and $\lambda_{rl} = 1$. However, setting $\lambda_{dis} = 1$ and $\lambda_{rl} = 0.5$ will not cause such problem. We speculate disentangling process brings better representation for L2I, generalizing model in a more efficient manner as correct samples' trajectory could be easily collected than models without disentangling.

**Reward stability**. We also check the reward stability during the RL process. As RL is introduced after the 10-th training epoch, we compute the averaged reward for sampled trajectories and visualize them in Fig. 6. From the figure, we conclude that although RL may encounter fluctuation during training, the training process is generally steady.

## 5 CONCLUSION

This paper advances layout to image (L2I) generation by addressing attribute leakages and generalization. For the first challenge, we devise instance disentanglement module (IDM) and utilize instance disentanglement constraint (IDC) for disentangling semantic-related features. IDC requires semantic-related features to yield more accurate attention maps than spurious ones, and thus avoids attribute leakage. For the second challenge, we formulate a reinforcement learning (RL) framework, enabling our method to learn with unpaired prompt-only data to improve model's generalization. These two contributions formulates our I-DRUID and achieves high accuracies on several benchmarks.

**Acknowledgments**. We acknowledge the support from Shanghai Municipal Commission of Economy and Informatization, under Grant No. 2024-GZL-RGZN-01008.

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

APPENDIX

---

**Algorithm 1** The Process of I-DRUID.

---

**Inputs:** Sample with $\mathbf{b}$ bounding boxes, a global prompt $p_g$ and $n$ instance captions $\mathbf{p}$. Reward model $r(\cdot)$, critic-net, actor-net, hyper-parameters $\lambda_{dis}$, $\lambda_{rl}$, training epochs $E$, action pool $\mathcal{MP}$, unpaired prompt $\{\mathbf{p}_u, \mathbf{b}_u, p_{g,u}\}$.

1: // Disentangling Stage.
2: Initialize L2I model with SD3-mid;
3: **for** $e$ in $E$ **do**
4:     Sample a batch of training data $\{\mathbf{b}_e, \mathbf{p}_e, p_g\}$;
5:     Sample time $t$;
6:     Obtain $\mathbf{R}^+$ with IDM and compute Eq. 4;
7:     Compute Eq. 11 with $\mathbf{R}^+$ and time $t$;
8:     // RL Stage.
9:     **if** $e \geq 10$ **then**;
10:         Sampling trajectory with unpaired data $\{\mathbf{p}_u, \mathbf{b}_u, p_{g,u}\}$ and enqueue actions into $\mathcal{MP}$ (only enqueue actions for the first 20% time steps);
11:         Obtaining generated images $X_0$;
12:         Obtaining reward with Eq. 6;
13:         Computing Eq. 7 and optimize critic-net;
14:         Computing advantage scores with Eq. 8;
15:         Sampling old actions of the same input from $\mathcal{MP}$;
16:         Obtaining policy for old and current actor;
17:         Computing Eq. 10 to obtain $L_{rl}$;
18:     **else**
19:         Setting $L_{rl}$ to 0;
20:     **end if**
21:     Optimizing L2I model through backpropagation with Eq. 12;
22: **end for**

---

**LLM Statement.** We use LLM to polish the writing, such as correcting grammar and other errors.

## A   OVERALL TRAINING PIPELINE

We demonstrate our overall training pipeline in Alg. 1. Our training contains two parts, *i.e.*, disentangling stage and RL stage. The former will be conducted for the whole training process to ensure basic L2I capability while the latter will be utilize only for the last 10 epochs to improve L2I model's generalization with unpaired data. It should be noted that action pool $\mathcal{MP}$ will be activated only for the last 10 epochs and will only save the actions for the first 20% time steps for faster RL. Moreover, during RL, actor- and critic-net will be optimized in a GAN-like manner. critic-net will be optimized 5 times before joining actor-net's optimization.

## B   IMPLEMENTATION DETAILS FOR UNET-BASED METHOD

Our method is also compatible with UNet-based diffusion models like SD-1.5. When transferring to SD-1.5, (1) IDM's location for deployment, (2) the objective for noise prediction, and (3) RL actions should be modified. For (1), we deploy IDM at the $8 \times 8$ mid-layer and $16 \times 16$ decoder layers of UNet to conduct feature disentanglement and subsequent cross attention. For other cross-attention layers without IDM, we optimize them with global prompt. For (2), as SD-1.5's training objective is noise prediction, $L_{ldm}$ should be changed (Rombach et al., 2022) accordingly. Finally, for (3), the actions for SD-1.5 follows original transitional probability formula at (Ho et al., 2020).

## C   FAILURE CASE STUDY

We visualize some failed cases in Fig. 7 to show its limitation. Specifically, our method is liable to generate flawed cases when small characters and human body emerges. However, the generation with

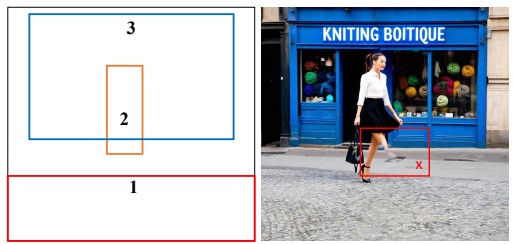 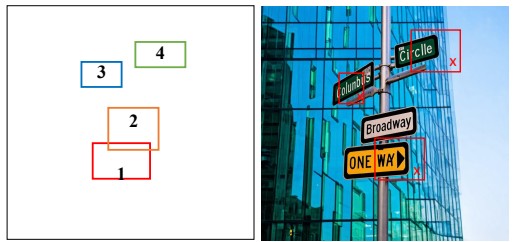

A [2]woman is walking on the [1]cobblestone street. Behind her is a blue shop front, with a [3]neon sign that reads 'KNITTING BOUTIQUE'.

Traffic signs that displays the street names [3-4]'Columbus Circle' and [2]'Broadway', as well as the traffic direction [1]'ONE WAY'.

Figure 7: Visualization of generated failure cases.

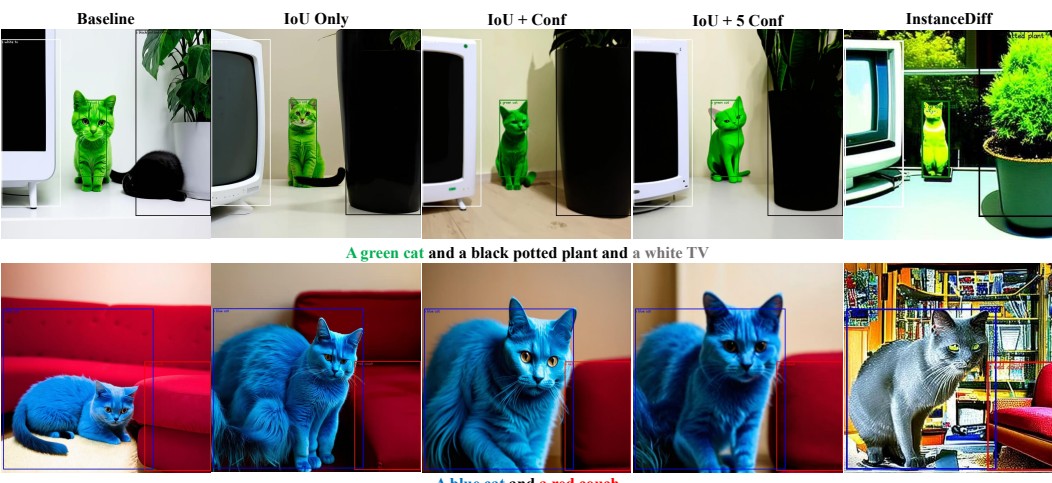

Figure 8: Justification of Reward Design.

small-scale text is a known issue in original stable diffusion (Esser et al., 2024), which will bring similar negative effects on all SD-based AIGC algorithms (Zhou et al., 2024b). To mitigate these negative effects, users could collect more data with text and human bodies to finetune the model.

We also note that previous SD1.5-based MIGC Zhou et al. (2024b) achieves better results than SD3-based Creati-Layout Zhang et al. (2025b) on COCO-MIG in terms of "Avg ISR". COCO-MIG is a benchmark highly focused on evaluating attribute leakage and rendering of each instance, suggesting weaker instance-level rendering capability for SD3-based L2I. We speculate the phenomenon is caused by the inherent problem in MM-attention, where text tokens are diluted in MM-attention Lv et al. (2025). Therefore, as MM-attention is heavily used in SD3, improving SD3-based L2I models generation capability in correctly rendering instance attributes becomes challenging. The problem further leads to the relatively lower L2I accuracies on COCO-MIG.

## D JUSTIFICATION OF REWARD DESIGN

As demonstrated in our experiments, we set the reward as "IoU + Confidence" to jointly consider spatial accuracies and instances' generation quality. To explore the optimal reward design, we conduct experiments by setting different weights to "Confidence" term, and visualize generated images for different variants.

Table 4: Quantitative Results for Reward Justification.

| ISR | L2 | L3 | L4 | L5 | L6 | Avg |
|---|---|---|---|---|---|---|
| Baseline | 63.30 | 61.04 | 57.46 | 52.20 | 50.10 | 56.82 |
| + IoU | 70.37 | 65.48 | 59.21 | 51.26 | 51.14 | 59.49 |
| + Conf | 76.87 | 69.16 | 62.96 | 52.37 | 52.39 | 62.75 |
| + 5Conf | 72.50 | 67.29 | 61.09 | 51.25 | 52.79 | 60.98 |

The results are shown in Fig. 8, we also visualize the results from InstanceD-iff Wang et al. (2024) for comparison. From the result, we draw two conclusions. (1) IoU term in RL enables better spatial control. Specifically, the "blue cat" at second row does not follow the given

Table 5: RL for initial states.

| Method | Avg ISR |
|---|---|
| Full-Time Sampling | 63.06 |
| Ours | 62.75 |

Table 6: Robustness to False Reward on COCO-MIG.

| ISR | L2 | L3 | L4 | L5 | L6 |
|---|---|---|---|---|---|
| w/o False Reward | 76.87 | 69.16 | 62.96 | 52.37 | 52.39 |
| w/ False Reward | 67.18 | 66.25 | 57.34 | 51.20 | 51.92 |

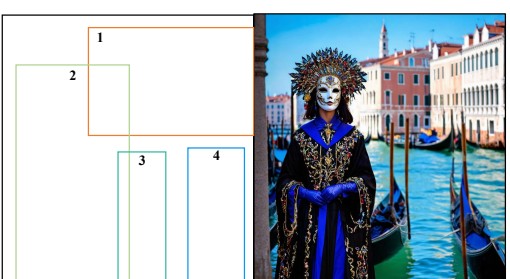

In the foreground, a figure dressed in exquisite traditional attire stands by the water. In the background, the iconic Venetian buildings and 3-4gondolas are clearly visible, with the water shimmering under the sunlight.

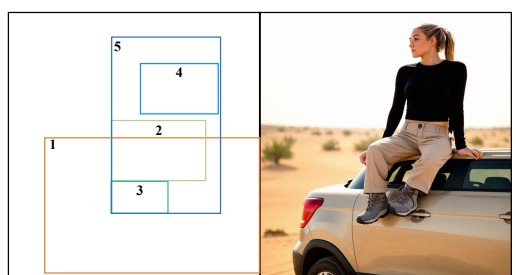

This is a photo of a woman sitting on the roof of a vehicle in a desert environment. The woman is dressed in a black top and beige trousers, wearing a pair of grey hiking shoes. She is sitting on the roof of a beige vehicle.

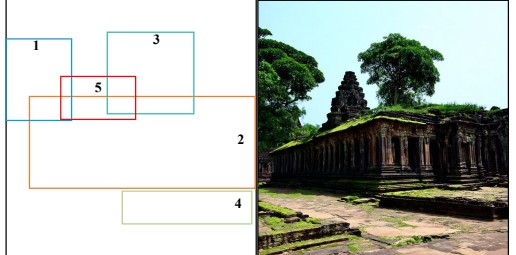

The temple is located in a tropical environment, with trees growing on its roof and walls. The walls of the temple are covered with moss. The ground is paved with stone slabs.

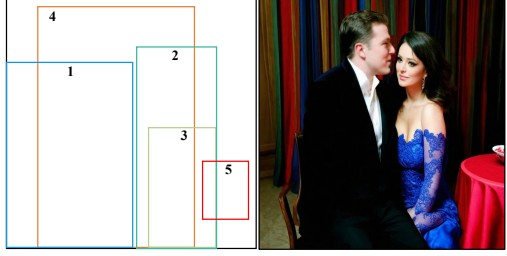

The man is dressed in a black suit, while the woman is wearing a blue evening gown with a lace design. They are sitting next to a table, with a curtain with red tablecloth in the background.

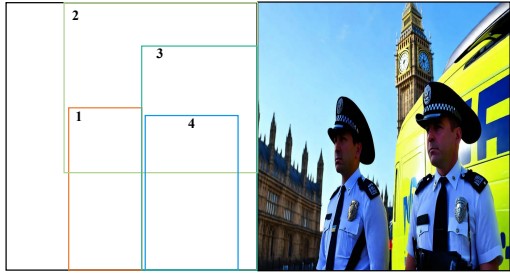

The officer on the left is wearing a black hat. The officer on the right is also wearing a black hat. They are standing in front of a yellow police van, with Big Ben and the surrounding buildings in the background

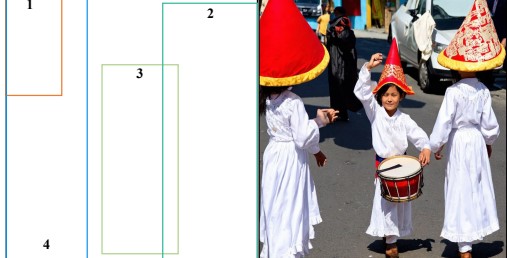

The child in the middle is holding a small drum in his hand. The left is a young child in festive white dress and oversized hat, while the right child is focused on her dancing.

Figure 9: Visualization of more cases (Part-I).

layout when RL is not applied. After using "IoU-based" RL, the spatial control is further improved. (2) Introducing confidence score into RL further improves instance-level generation, but may lead to low image quality. Specifically, ignoring "confidence term" will lead to the unnatural generation of some instances (e.g., green cat with a black tail, or blue cat with multiple legs). These artifacts are alleviated after taking "confidence term" into RL. However, when changing the reward function to "IoU + 5 * confidence", we note over-saturation for some generated instances. We thus adopt "IoU + confidence" as our final reward. The quantitative results on COCO-MIG at Tab. 4 demonstrate the same conclusion.

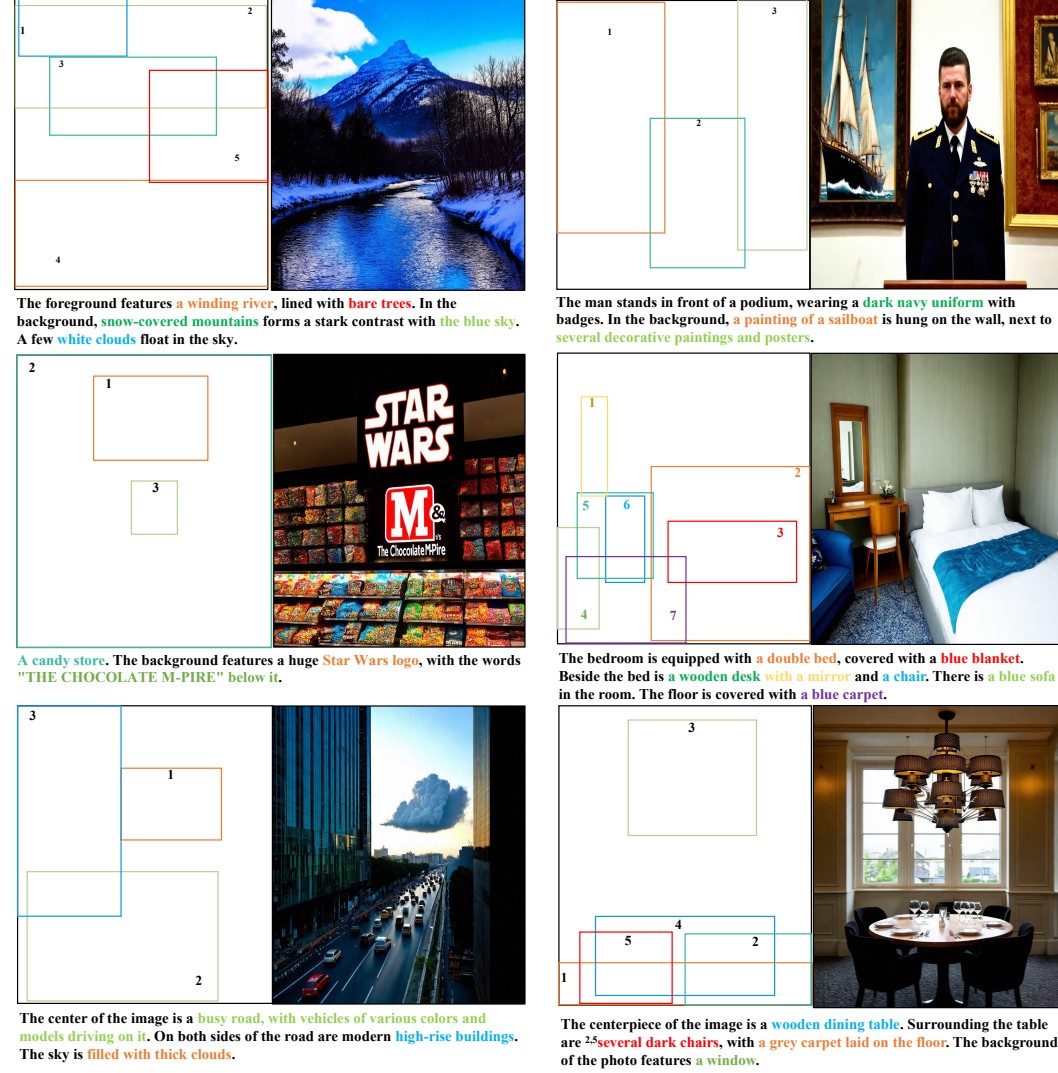

Figure 10: Visualization of more cases (Part-II).

# E   RL AT EARLY TIME STEPS

We only conduct RL on the first $20\%$ time steps to speed up the training process, raising the concern whether RL on initial states yields similar improvement on L2I trajectory sampling with full time steps (we term this variant as full-time sampling). We conduct experiment with full time trajectory sampling and evaluate L2I model on COCO-MIG to demonstrate this. As shown in Tab. 5, using full-time sampling yields slight improvement but requires higher computational cost as optimization will be conducted on the whole trajectory. Specifically, with RL training at early time steps, our method achieves training speed of nearly 15.4s / iter on NVIDIA H20, while the training speed with full trajectory sampling is nearly 82.6 s / iter. We thus propose to optimize initial states for faster RL.

# F   MORE VISUALIZATION

We visualize more results in Fig. 9 and Fig. 10 to show our method's efficacy in L2I. Specifically, our method achieves both spatial correctness and high instance image quality, demonstrating its potential in L2I field.

# G ROBUSTNESS TO UNRELIABLE REWARD

We also conduct experiment to check our method's robustness when encountering false reward by deliberately assigning false reward (minus of the reward). The results are shown in Tab. 6.

From the result, we conclude that our RL part suffers from performance degradation when wrong rewards are given, but the performance degradation does not cause catastrophic false generation. This is due to two reasons:

(1) Joint optimization with paired data and KL terms. This is a commonly used strategy to overcome training instability during RL Stiennon et al. (2020); Fan et al. (2023); Black et al. (2023). By using KL term and keeping original pair-wise training loss, our method mitigates the negative effects brought by RL. Moreover, we only conduct RL at the last 10 epochs of training, where our model already has strong capability for L2I generation. During the trajectory sampling, the generated images will not introduce samples with completely false layout, alleviating the negative effects brought by false reward.

(2) Feature Disentanglement. As demonstrated in Tab. 3, after using IDM for RL, the final L2I accuracies are improved. Therefore, our IDM provides better representation for subsequent RL.

To further alleviate the negative effects brought by false reward, one can combine multiple reward functions (combining multiple detectors, *etc.*) to improve reward robustness, or carefully adjusting KL weights to strike a balance between aggressive exploration or stable training.

# H PSEUDO CODE OF OUR I-DRUID

We provide the pytorch-like pseudo code for our disentangling module and RL training. Specifically, "CAS score computation" is the function to compute CAS score. "Compute $L_{dis}$" is the function to compute $L_{dis}$, "Compute $L_{rl}$" is the function to compute $L_{rl}$, while "SDE-ODE Conversion" is conducted on original "FlowMatchingScheduler" of SD3 to implement Eq. 5.

CAS score computation

```python
def _get_bg_attn_loss(self, worse_attn, bg_mask):
    BPN, heads, HW, _ = ori_attnmap.shape
    B = bg_mask.shape[0]
    bg_mask = bg_mask.reshape(-1, 1, HW)

    # get ori bg attn map
    ori_attnmap = torch.sum(ori_attnmap[:, :, :, 1:], dim=-1)
    ori_attnmap = ori_attnmap.reshape(B, -1, HW)
    bg_mask = bg_mask.reshape(B, 1, HW)

    # get general bg info
    bg_attn_mean = (ori_attnmap * bg_mask).sum(dim=-1) / \
        ((bg_mask).sum(dim=-1) + 1e-6) # (B, PN*heads)
    loss_attn = (abs(ori_attnmap - bg_attn_mean[..., None].detach()) \
        * bg_mask).sum(dim=-1) / (bg_mask.sum(dim=-1) + 1e-6)
    return loss_attn.mean()
```

Compute $L_{dis}$

```python
def _get_disen_loss(self, worse_attn, better_attn, masks):
    ranking_loss = torch.nn.SoftMarginLoss()
    H_ori = self._get_bg_attn_loss(worse_attn, masks)
    H_better = self._get_bg_attn_loss(better_attn, masks)
    y = torch.ones_like(H_ori)
    return ranking_loss(H_ori-H_better, y)
```

Compute $L_{rl}$

```python
def rl_train(self, state_dict, chosen_pid, height, width, \
    clip_range=1e-4):
    cur_glob = [state_dict["caption"][pid] for pid in chosen_pid]
    sub_prts = [state_dict["sub_prompts"][pid] for pid in chosen_pid]
    sub_bbox = [state_dict["bboxes"][pid] for pid in chosen_pid]
    ref_logp = state_dict["log_prob"][chosen_pid, :]
    cur_logp, cur_reward, infer_img = self.pipe.train_get_logp(
        cur_glob, height, width, num_inference_steps=50, \
        max_objs = 10, timesteps = None,
        bbox_phrases=sub_prts, bbox_raw=sub_bbox, \
        reward_func=self.reward_func,
        load_image=self.load_image, topk=10
    )
    topk = ref_logp.shape[-1]
    cur_adv = cur_reward.to(cur_logp.device).view(-1,1) -\
        self.value_func(infer_img)

    ratio = torch.exp(cur_logp[:, :topk] - \
        ref_logp.to(cur_logp.device))
    unclipped_loss = -cur_adv * ratio
    clipped_loss = -cur_adv * torch.clamp(
        ratio, 1.0 - clip_range, 1.0 + clip_range,
    )
    return torch.mean(torch.maximum(unclipped_loss, clipped_loss)) +\
        self.kl_loss(cur_logp, ref_logp)
```

ODE-SDE Conversion

```python
def sde_step_with_logprob(
    self, model_output, timestep, sample = 0.7,
    prev_sample = None, generator = None,
):
    sample=sample.float()
    if prev_sample is not None:
        prev_sample=prev_sample.float()

    step_index = [self.index_for_timestep(t) for t in timestep]
    prev_step_index = [step+1 for step in step_index]
    sigma = self.sigmas[step_index].view(
        -1, *([1] * (len(sample.shape) - 1))
    )
    sigma_prev = self.sigmas[prev_step_index].view(
        -1, *([1] * (len(sample.shape) - 1))
    )
    sigma_max = self.sigmas[1].item()
    dt = sigma_prev - sigma
    std_dev_t = torch.sqrt(sigma / \
        (1 - torch.where(sigma == 1, sigma_max, sigma)))*noise_level

    # sde
    prev_sample_mean = sample* (1+std_dev_t**2/(2*sigma)*dt)+\
        model_output*(1+std_dev_t**2*(1-sigma)/(2*sigma))*dt

    if prev_sample is None:
        variance_noise = randn_tensor(
            model_output.shape,
            generator=generator,
            device=model_output.device,
            dtype=model_output.dtype,
        )
        prev_sample = prev_sample_mean +
            std_dev_t * torch.sqrt(-1*dt) * variance_noise
```

```python
    log_prob = (
        -((prev_sample.detach() - prev_sample_mean) ** 2) /
        (2 * ((std_dev_t * torch.sqrt(-1*dt))**2))
        - torch.log(std_dev_t * torch.sqrt(-1*dt))
        - torch.log(torch.sqrt(2 * torch.as_tensor(math.pi)))
    )

    log_prob = log_prob.mean(dim=tuple(range(1,
                log_prob.ndim)))
    return prev_sample, log_prob, prev_sample_mean, std_dev_t

def compute_log_prob(self, noise_pred, sample, time):
    prev_sample, log_prob, prev_sample_mean, std_dev_t =
    self.sde_step_with_logprob(
        noise_pred.float(), time, sample.float(),
        prev_sample=None, noise_level=0.7,
    )
    return prev_sample, log_prob, prev_sample_mean, std_dev_t
```

