# OpenReview forum: "I-DRUID: Layout to image generation via instance-disentangled representation and unpaired data"
_ICLR.cc/2026/Conference — ICLR 2026 Poster_

### Official Review · Reviewer_uTfS · 2025-10-29

**Soundness:** 3
**Presentation:** 3
**Contribution:** 3
**Rating:** 6
**Confidence:** 3

**Summary:**

This paper targets for solving two issues in L2I generation, including instance overlapping and poor generation to new scenes. The method it proposed has two components, 1/ the instance-disentanglement module to disentangle semantic-related features and spurious features; 2/ RL module to generate unpaired data for model training. The evaluation on COCO_MIG and LayoutSAM-eval demonstrates that the proposed method achieves the SOTA performance compared with prior work.

**Strengths:**

- The way of incorporating RL on unpaired data into L2I model training is impressive, which indeed solves the generation limitation of current L2I approaches. The conversion to SDE solves an emerging pain point of exploring flow-matching models.

- The ablation and sensitivity analyses make the paper solid, especially the illustration of module effectiveness.

- The visualization is also good to follow with both good cases and failure cases provided.

**Weaknesses:**

- IDC is heuristic and does not stand out compared to previous attention ideas. The novelty in CAS-based Softplus inequality is incremental to existing attention regularization in L2I, such as region-aware attention, for example, Rethinking The Training And Evaluation of Rich-Context Layout-to-Image Generation (NeuIPS 2024), which should also be compared in the result tables.

- The reward design (GDINO IoU plus confidence) mainly measures the spatial grounding, ignoring the attribute binding such as color and texture, and text fidelity. This should be improved in the future.

- Comparison fairness: the results mix SD-1.5 and SD-3 while some baselines are UNET-based only. It would be better to use stronger base model such as SD-3 for the baselines if possible.

- The contents are too long. The introduction of experimental settings should be included in the original paper rather than appendix.

**Questions:**

- Why SD-1.5 has better performance than SD-3 on COCO-MIG and is it because IDM helps one backbone more than the other?

- Is there any overlap or duplicate risk for testing on COCO-MIG since RL uses COCO-2014-MIG prompts for generating unpaired data?

---

> ### Author Response · Authors · 2025-11-21
> **Response to Reviewer uTfS**
>
> Thanks for acknowledging our way of incorporating RL into training with unpaired data.
>
> **Q1**. Comparison with "Rethinking The Training And Evaluation of Rich-Context Layout-to-Image Generation".
>
> **A1**. "RichContext" is a pioneering work to achieve L2I through fine-level manipulation on cross-attention map. We evaluate it on COCO-MIG and report the results in Tab.1 of the revision. RichContext's Avg ISR is 30.76 and Avg mIoU is 30.93. The results demonstrates that our method outperforms theirs on this benchmark. We have added the result in the revision.
>
>  **Q2**. The reward design mainly measures the spatial grounding, ignoring the attribute binding such as color and texture, and text fidelity. This should be improved in the future.
>
> **A2**. Thanks for your advice. We used to add other metrics (CLIP scores, *etc.*) to enrich our reward design but they brought nearly no improvement. We leave the optimal design of reward function as a future work.
>
> **Q3**. Comparison fairness: the results mix SD-1.5 and SD-3 while some baselines are UNET-based only. It would be better to use stronger base model such as SD-3 for the baselines if possible.
>
> **A3**. Thanks. We have reported and compared the state-of-art SD3-based L2I work (Creati-Layout) in Tab.1 and 2. However, we admit that the development of SD3-based L2I models is ongoing and few SD3-based L2I methods could be involved for fair comparison. We therefore combine all previous (both SD1.5-based and SD3-based) methods and show our method's flexibility.
>
> **Q4**. The contents are too long. The introduction of experimental settings should be included in the original paper rather than appendix.
>
> **A4**. Thanks, we have moved the experimental specifications to main paper in the revision.
>
> **Q5**. Why SD-1.5 performs better than SD3 on COCO-MIG?
>
> **A5**. That is a good observation. We also note that previous SD1.5-based MIGC achieves better results than Creati-Layout on COCO-MIG in terms of Avg ISR. COCO-MIG is a benchmark highly focused on evaluating attribute leakage and rendering of each instance, suggesting weaker instance-level rendering capability for SD3-based L2I. We speculate the phenomenon is caused by the inherent problem in MM-attention, where text tokens are diluted in MM-attention [a]. Therefore, as MM-attention is heavily used in SD3, improving SD3-based L2I models' generation capability becomes challenging. The problem further leads to the relatively lower L2I accuracies on COCO-MIG. We have added the discussion in the revision, at Appendix--Sec.C: "Failure Case Study".
>
> **Q6**. Is there any overlap or duplicate risk for testing on COCO-MIG since RL uses COCO-2014-MIG prompts for generating unpaired data?
>
> **A6**. We would like to emphasize that the RL training is conducted by using prompt-only data. The images are not included in the training. Therefore, this is not considered to be a duplicate risk for testing.
>
> [a] Rethinking Cross-Modal Interaction in Multimodal Diffusion Transformers. In ICCV'25.

---

> ### Comment · Reviewer_uTfS · 2025-11-27
>
> Thanks the authors for addressing most of my concerns. The quantitative comparisons in the revision are now more fair, and the discussion provides a more comprehensive understanding of the evaluation results on COCO-MIG. Overall, I am inclined to recommend acceptance of this work.

---

### Official Review · Reviewer_6mc3 · 2025-10-29

**Soundness:** 3
**Presentation:** 2
**Contribution:** 2
**Rating:** 4
**Confidence:** 4

**Summary:**

This work proposes to use the feature disentanglement and reinforcement learning to improve the performance of layout-to-image generation task. The author designs an instance disentanglement module to separate the image feature and encourages the model to concentrate semantic-related information into R+ while “spurious parts” into R-. Afterwards, this work uses the reinforcement learning with PPO to train the model. The final model is optimized jointly with the diffusion loss, disentanglement loss and RL loss.

**Strengths:**

* The motivation to separate the semantic-related features and spurious features is interesting.

**Weaknesses:**

* From the abstract, it seems that the purpose of feature disentanglement is for avoiding attribute leakage, is there any evidence that support that attribute leakage has been mitigated after using the feature disentanglement?
* The reviewer would like the author to clarify  the novelty in the RL part. From the current manuscript, it seems that the method just adopts the PPO and uses a new reward for layout-to-image task. Also I would like the author to quantitatively justify their choice of reward, e.g., why use IoU + confidence, what if change the importance ratio between them.
* The reviewer has several questions for instance disentanglement part.
    * ln 196, what does “global-prompt-enhanced features” mean?
    * if the reviewer understands correctly, the purpose of the disentanglement is to concentrate the semantic-related features in cross-attention region, i.e., increasing the response in R+ while decreasing the response in R-. While from the illustration in Fig 2, it seems that in both building and background regions, the R- has higher response. The reviewer wonders if this observation suggests that the disentanglement is not work as expected.
    * the reviewer wonders the design rational of Eq 3. the (1-M) part is understandable as noted in ln 218 (evaluates the degree of attention value beyond instance i’s scope.) but why use R+ - AVG(R+)? What is the purpose of computing the difference between each instance response and average response? And what is the point to have the average response?
    * ln 225-227 mentioned that “Minimizing the above disentanglement loss Ldis enables IDM to accurately separate each instance and thus avoid attribute leakage explicitly.”. why minimizing Eq 4 can lead to instance separation and avoid attribute leakage?

**Questions:**

Please see my weaknesses

---

> ### Author Response · Authors · 2025-11-21
> **Response to Reviewer 6mc3 (Part-I)**
>
> Thanks for acknowledging our contribution in separating the semantic-related features and spurious features for L2I. Our point-to-point response are as follow.
>
> **Q1**. The evidence of alleviating attribute leakage ?
>
> **A1**. The evidence of alleviating such problem is two-fold.
>
> (1) Improvement on COCO-MIG and LayoutSAM-eval in terms of "ISR" and "color" metrics. Specifically, COCO-MIG [b] is originally designed to evaluate L2I model's capability in attribute leakage as each instance of testing prompts are deliberately assigned with a color attribute. For the generated image, COCO-MIG check each instance's color attribute. If one of the instances are considered to be wrong, the whole sample is considered to be a failure. Therefore, the improvement over "Avg ISR" between "No. 1" and "No. 2" of Tab.3 demonstrates that attribute leakage has been alleviated.
>
> (2) Visualization. We add more visualization at ablation study of revised manuscript to show each component's direct effect on generated images. Specifically, in our revised manuscript, noticeable attribute leakage could be found when solely using $L_{ldm}$ for optimization (at the column of "$L_{ldm}$"). After introducing $L_{dis}$, the attribute leakage is alleviated. Moreover, adding $L_{rl}$ achieves fine-level attribute control for instances. Please find these results in our revision, at Sec. 4.4--"Ablated Visual Comparison".
>
> To sum up, we demonstrate our model's efficacy in handling attribute leakage in terms of both quantitative metrics and visual results. We hope these two points will make the reviewer realize the necessity of feature disentanglement.
>
> **Q2**. The reviewer would like the author to clarify the novelty in the RL part.
>
> **A2**. We would like to further emphasize the novelty of our RL part.
>
> (1) RL is our main strategy to achieve training with unpaired, prompt-only data, which is less explored in previous L2I works. Specifically, previous L2I works [a,b,c] collect image-text pairs with VLMs and fine-tune their model on the collected dataset. To the best of our knowledge, we are the first L2I method that integrates unpaired, prompt-only data into the training process.
>
> (2) To speed up RL, we propose to conduct RL only at the early time steps for the obtained generation trajectories. As demonstrated in appendix--Sec. E, our method improves RL training efficiency while keeping good L2I capability.
>
> **Q3**. Also I would like the author to quantitatively justify their choice of reward.
>
> **A3**. Thanks. We add more visualizations to show the rationale behind our reward design. Specifically, we set different weights to "confidence" term, and visualize generated images for different variants. The visualizations are shown in our revision (at Appendix--Sec. D: "Justification of Reward Design"). From the result, we draw two conclusions.
>
> (1) **IoU term in RL enables better spatial control**. Specifically, the "blue cat" at second row does not follow the given layout when RL is not applied. After using "IoU-only" RL, the spatial control is improved.
>
> (2) **Introducing confidence score into RL further improves instance-level generation, but assigning high weight on it may lead to low image quality**. Specifically, ignoring "confidence term" will lead to the unnatural generation of some instances. These artifacts are alleviated after taking "confidence term" into RL. However, when changing the reward function to "IoU + 5 confidence", we note over-saturation for some generated instances. We therefore adopt "IoU + confidence" as our final reward.

---

> ### Author Response · Authors · 2025-11-21
> **Response to Reviewer 6mc3 (Part-II)**
>
> **Q4**. The reviewer has several questions for instance disentanglement part. (1) In 196, what does "global-prompt-enhanced features" mean? (2) from the illustration in Fig 2, it seems that in both building and background regions, the R- has higher response. The reviewer wonders if this observation suggests that the disentanglement is not work as expected. (3) why use R+ - AVG(R+) in eq.3 ? (4) why minimizing Eq 4 can lead to instance separation and avoid attribute leakage?
>
> **A4**. Thanks. We would like to explain each part of our design of disentanglement loss.
>
> (1) It is the image feature after MM-attention with global prompt. We are sorry for the confusion and have corrected the overall flowchart at Fig. 2.
>
> (2) Eq. 3 is calculated specifically over the background region $(\mathbf{1}-\mathcal{M}_{i})$ and measures the "degree of attention value beyond instance i's scope". Therefore, $\mathbf{R}^{+}$ should trigger lower response than $\mathbf{R}^{-}$ beyond the instances and Fig.2's feature map (triggered with instance prompt "Historical tower") well-align this design.
>
> (3) First, we would like to further explain that AVG($\cdot$) is a channel-wise operation. For the given $\mathbf{R}^{+}$ with size of $B \times C \times H \times W$, AVG($\cdot$) converts it to $B \times 1 \times H \times W$. Based on this clarification, Eq.3 measures the degree of attention value beyond instance i's scope while restricting the response close to the background's averaged level. We have clarified this in the revision.
>
> (4) Minimizing Eq.4 is equivalent to encourage lower CAS score for $\mathbf{R}^{+}$ and higher CAS for $\mathbf{R}^{-}$, thus enabling our disentanglement to heuristically learn semantic-related features for L2I generation.
>
> [a] Creati-Layout: Siamese Multimodal Diffusion Transformer for Creative Layout-to-Image Generation. In ICCV'25.
>
> [b] MIGC: Multi-Instance Generation Controller for Text-to-Image Synthesis. In CVPR'24.
>
> [c] InstanceDiffusion: Instance-level Control for Image Generation. In CVPR'24.

---

> > ### Comment · Reviewer_6mc3 · 2025-11-27
> >
> > The reviewer appreciates author's response. While some of my concerns have been addressed, some of my concerns remains and the newly added contents raise additional questions. Therefore, the reviewer would like to ask several follow up questions.
> >
> > Regarding the novelty in the RL part. The reviewer is not satisfied with the current response. While it might be true that the L2I community have not applied the RL for this generative task. However, the RL has been widely used in the generative diffusion model community, and L2I tasks is a de facto branch of the genrative diffusion models. The proposed RL in current manuscript seems more like an engineering adaption to L2I by adding layout condition but the RL strategy itself is kept as it is.
> >
> > Regarding quantitative justification of the choice of reward. The reviewer has read Sec D in the updated manuscript. Two concerns are raised:
> > 1. The provided result is a qualitative comparison, there is no quantitative evidence to support the default choice of the IoU+Conf scheme.
> > 2. The visualization results raise the reviewer concern on the effectiveness of the proposed method. It seems that the bounding box is not tight at all. The reviewer has worked with one of the baseline InstanceDiffusion a lot and this baseline method generates objects tightly aligning with the given bounding box. However, the current method often produces results that is either loose in the bbox or beyond the boundary of the bbox (in figure 8, right three columns). This raises a question on the reliability of the "spatial" evaluation in the Table 2 and 3. Moreover, according to author's response, "assigning high weight (to conf term) on it may lead to low image quality", which in other word the reviewer would infer that the layout fidelity is more controlled by IoU in the reward. However, the second column of the "blue cat" example does not seem to support this conjugation. In this example, the reward is solely composed of IoU, but the red couch is not well aligned with given bounding box.
> >
> > Regarding instance disentanglement.
> >
> > Applying AVG across the channel dimension is even more confusing, what is the point to compute the channel-wise average and then computing each channel's different to the mean channel value? Using (1-M) to specify the non-instance region is understandable but it is not convincing enough to say this difference "measures the degree of attention value beyond instance i's scope". If the purpose is meansuring the degree of attention value, there is no point to substract the AVG feature map.
> >
> > And this operation is also applied on the spurious part R-, the current claimed purpose of Eq 3 is for R+, what would be the Eq 3's meaning when it is applied on the R-?

---

> > > ### Author Response · Authors · 2025-11-29
> > > **Response to further questions (Part-I)**
> > >
> > > **Q5**. The proposed RL in current manuscript seems more like an engineering adaption to L2I by adding layout condition but the RL strategy itself is kept as it is.
> > >
> > > **A5**. We thank the reviewer for providing further response regarding the novelty of our method, which the reviewer characterizes as an "engineering adaptation". We respectfully disagree and wish to clarify our methodological contribution based on two key aspects: Technology and Motivation.
> > >
> > > **Technology**. Our method proposes to conduct RL on semantic-related policy space, which is proven to be a better policy space than standard feature space for L2I. Although RL itself is based on previous works, conducting RL on semantic-related feature space while obtaining such features with our IDM is original. Moreover, as demonstrated in our experiment and response to Reviewer WrbM's Q1-A1, simply applying RL to L2I without disentanglement achieves minor improvement, which further demonstrates the necessity of our technology.
> > >
> > > **Motivation**. Our method is motivated by the need to enhance generalization with unpaired, prompt-only data, which is under-explored in concurrent works and acknowledged by Reviewer uTfS and h5RF.
> > >
> > > By clarifying the necessity of our technology and the innovative motivation behind utilizing unpaired data, we hope to demonstrate the non-trivial design and novelty of our method. We sincerely appreciate reviewer h5RF, WrbM, and uTfS in acknowledging our novelty and contributions. We would also like to kindly ask Reviewer 6mc3 to reconsider the non-trivial design of our framework.
> > >
> > > **Q6**. The provided result is a qualitative comparison, there is no quantitative evidence to support the default choice of the IoU+Conf scheme.
> > >
> > > **A6**. We provide quantitative metrics as follow:
> > >
> > > | ISR | L2 | L3 | L4 | L5 | L6 | Avg |
> > > | :---: | :---: | :---: | :---: | :---: | :---: | :---: |
> > > | Baseline | 63.30 | 61.04 | 57.46 | 52.20 | 50.10 | 56.82 |
> > > | IoU | 70.37 | 65.48 | 59.21 | 51.26 | 51.14 |  59.49 |
> > > | IoU + Conf | 76.87 | 69.16 | 62.96 | 52.37 | 52.39 | 62.75 |
> > > | IoU + 5 Conf  | 72.50 | 67.29 | 61.09 | 51.25 | 52.79 | 60.98 |
> > >
> > > The quantitative results further support our speculation. i.e., IoU term is beneficial to spatial control, while IoU + Conf  further improves instance generation (the improvement also includes spatial control).
> > >
> > > **Q7.** (1) It seems that the bounding box is not tight at all. (2) Moreover, according to author's response, "assigning high weight (to conf term) on it may lead to low image quality", which in other word the reviewer would infer that the layout fidelity is more controlled by IoU in the reward. However, the second column of the "blue cat" example does not seem to support this conjugation. In this example, the reward is solely composed of IoU, but the red couch is not well aligned with given bounding box.
> > >
> > > **A7.**  We thank reviewer's careful observation. (1) We add the generated results with InstanceDiff in Fig. 8 as the reference images. We find that InstanceDiff also encounters layout misalignment on these two cases (layout misalignment for ''blue cat'' and ''black potted plant'') while some generated instances present low quality or incorrect attribute (e.g., the over-saturation for the green cat and incorrect color for "blue cat''). Compared to InstanceDiff, our method presents better image quality and spatial control. (2)  We do not deny the effectiveness of confidence term on spatial control, we claim that "confidence term further improves instance-level generation'' which inherently improves spatial control. The quantitive results at Q6-A6 also support this conclusion. For the "red couch" case, the spatial control of red couch is further improved after taking confidence term into optimization, which aligns with our conclusion.
> > >
> > > **Q8**. Applying AVG across the channel dimension is even more confusing.
> > >
> > > **A8**. Thanks for the reviewer's detailed response. We indeed miswrite Eq.3. We have revised our manuscript and would like to explain the rationale behind our loss design. Specifically, Eq.3 should be:
> > >
> > > CAS(R_CA+, M) = Sum_{i=1}^{n} | R_{CA,i}^{+} - AVG(R_{CA,i}^{+} * (1 - M_i)) | * (1 - M_i)
> > >
> > > where M_{i} and R_{CA,i}^{+} are the mask and triggered attention map for instance i, respectively. In our design, AVG(R_{CA,i}^{+} * (1 - M_i)) extracts instance i's averaged background attention values. Eq.3 subsequently quantifies the total absolute deviation beyond instance i's scope. Higher CAS indicates potential bounding box mis-alignment for the generated instance as the leaked instance will lead to high dispersion at background. Please check our revision.

---

> > > > ### Author Response · Authors · 2025-11-29
> > > > **Response to further questions (Part-II)**
> > > >
> > > > **Q9**.  And this operation is also applied on the spurious part R-, the current claimed purpose of Eq 3 is for R+, what would be the Eq 3's meaning when it is applied on the R-?
> > > >
> > > > **A9**. Based on A8, higher CAS indicates potential bounding box mis-alignment for the generated instance as the instance will lead to high dispersion at background. Therefore, we minimize Eq.4, encouraging IDM to heuristically seek for R- that lead to failure of instance generation (i.e., high CAS scores), while finding R+ to achieve better L2I capability  (i.e., lower CAS scores). R- will be discarded to achieve better L2I capability.

---

### Official Review · Reviewer_h5RF · 2025-11-02

**Soundness:** 3
**Presentation:** 3
**Contribution:** 3
**Rating:** 6
**Confidence:** 4

**Summary:**

The paper proposes I-DRUID, a framework for layout-to-image (L2I) generation that combines reinforcement learning and instance disentanglement learning to enhance generalization and spatial consistency. The proposed method eliminates attribute leakage between instances by introducing the Instance Disentanglement Module (IDM) and Instance Disentanglement Constraint (IDC), which separate semantically relevant and irrelevant features. A Reinforcement Learning (RL) module is applied on top of a diffusion-based generator, trained using PPO optimization on unpaired prompt-only data and guided by feedback from an external grounding model (Grounding-DINO).

**Strengths:**

+ Original combination of disentanglement and RL in diffusion-based L2I generation.
+ Experimental results demonstrated improved spatial alignment and instance fidelity.
+ The use of unpaired data is practical and reduces annotation costs.
+ Ablation studies show evidence of reduced attribute leakage.

**Weaknesses:**

- It is questionable to employ Grounding-DINO as a reward evaluator. Although it excels at object detection, it was trained for spatial grounding on real images rather than perceptual or stylistic quality. As a result, the reward may be biased toward positional correctness, unstable on synthetic generations, and insensitive to visual realism, leading to inconsistent RL behavior across datasets.
- No quantitative analysis of computational cost, which is critical given the RL overhead.
- Evaluation metrics mostly focus on geometric accuracy (mIoU, ISR); perceptual or semantic quality is not deeply discussed.

**Questions:**

1. Inconsistency between SD-1.5 and SD-3 results across Tables 1 and 2. The reported results are not consistent between the two backbones. On COCO-MIG (Table 1), SD-1.5 clearly outperforms SD-3 with higher Instance Success Rate (69.13 vs 62.75) and mIoU (68.18 vs 55.60). However, on LayoutSAM-eval (Table 2), the trend reverses, SD-3 achieves markedly better scores (Spatial 93.14 vs 86.95, Color 75.37 vs 70.49, Texture 78.35 vs 73.56, Shape 77.20 vs 72.30, FID 17.21 vs 22.92, IS 22.45 vs 17.95). Could the authors clarify why?
2. Grounding-DINO is primarily a detection model trained on real images, focusing on spatial grounding rather than perceptual or stylistic quality. When used as a reward function for diffusion RL, its outputs may be biased toward positional correctness while ignoring visual realism and may also be unreliable on synthetic generations. How do the authors mitigate these biases or ensure that the reward signal remains stable and meaningful during RL training?
3. Could the authors provide runtime or FLOPs comparison to quantify efficiency versus baselines?

**Details Of Ethics Concerns:**

N.A.

---

> ### Author Response · Authors · 2025-11-21
> **Response to Reviewer h5RF**
>
> Thanks for acknowledging our contribution to L2I in terms of creativity and experiments. Our point-to-point response are as follow.
>
> **Q1**. On COCO-MIG, SD-1.5 clearly outperforms SD-3 with higher ISR and mIoU. However, on LayoutSAM-eval, the trend reverses. Could the authors clarify why?
>
> **A1**. We would like to further clarify this. We first explain why SD3-based variant outperforms SD1.5-variant on LayoutSAM-eval. Different from SD1.5 that interprets prompt with CLIP and has 77-token limit, SD3 introduces T5 text-encoder to handle longer captions. As LayoutSAM-eval has longer averaged testing prompts than COCO-MIG (95.41 tokens [a]), it is inevitable to avoid prompt truncation at inference, leading to their poorer performance than SD3-based methods.
>
> We then explain why SD3-variant performs poorer on COCO-MIG than SD1.5 variant. This is the same concern as Reviewer uTfS's Q5, which is caused by inherent flaw of MM-attention. Please refer to Reviewer uTfS's Q5-A5 for more details.
>
> To sum up, SD1.5 performs better on testing prompt with short captions than SD3 due to inherent flaw of MM-attention, but SD3-based L2I has advantage on testing cases with long-captions as 77-token limit is alleviated.
>
> **Q2**. How do the authors ensure that the reward signal remains stable and meaningful during RL training ?
>
> **A2**. Thanks. This is the same concern as Reviewer WrbM's Q1. During the rebuttal, we conduct experiments when false reward scores are given for RL to check method's robustness. Our method indeed suffers from performance degradation due to false reward but the performance drop is not very significant due to "Joint optimization with paired data", "KL regularization", and "Do not introduce RL at early epochs". Please refer to Reviewer WrbM's Q1-A1 for more details.
>
> **Q3**. Could the authors provide runtime comparison to quantify efficiency versus baselines ?
>
> **A3**. Thanks. We have already report runtime in the revised manuscript, at L\#358, "our method takes 4 days on 8 NVIDIA H20 GPUs for training". As a comparison, Creati-Layout will take 7 days on 8 A800-40G GPUs [a], which is equivalent to 3.5 days on 8 NVIDIA H20 GPUs. Therefore, our method is nearly as efficient as Creati-Layout. Please refer to our revision.
>
> **Q4**. Evaluation metrics mostly focus on geometric accuracy (mIoU, ISR); perceptual or semantic quality is not deeply discussed.
>
> **A4**. We have reported perceptual or semantic quality (FID, PickScore, IS) at Tab.2. Please find them in our revision.
>
> [a] Creati-Layout: Siamese Multimodal Diffusion Transformer for Creative Layout-to-Image Generation. In ICCV'25.

---

### Official Review · Reviewer_WrbM · 2025-11-02

**Soundness:** 4
**Presentation:** 4
**Contribution:** 3
**Rating:** 4
**Confidence:** 5

**Summary:**

This paper introduces I-DRUID, a novel framework for Layout-to-Image (L2I) generation that addresses the critical challenges of attribute leakage and limited generalization. To prevent attribute leakage between instances, it proposes an Instance Disentanglement Module (IDM) that isolates semantic features for more precise attention control. To enhance generalization to novel scenes, the framework uniquely leverages reinforcement learning (RL) with unpaired, prompt-only data, enabling the model to learn from diverse AI feedback without requiring expensive paired image-text datasets. The method demonstrates state-of-the-art performance across multiple benchmarks and shows high flexibility by successfully adapting to both UNet and MM-DiT architectures.

**Strengths:**

1. The paper introduces a creative combination of instance-level disentanglement and reinforcement learning (RL) to address challenges in layout-to-image (L2I) generation.
2. The paper is logically structured and the language is fluent.
3. A comprehensive performance comparison was conducted against several recent SOTA methods, achieving leading results on multiple metrics and fully demonstrating the superiority of the proposed approach.

**Weaknesses:**

1. Using Grounding-DINO as a fixed reward model introduces bias and limits generalization. The paper does not examine robustness when GDINO fails to detect objects or misinterprets captions.
2. The framework is too complex and maks this work hard to follow. The I-DRUID framework integrates multiple complex components. While its performance is superior, its high complexity may pose challenges for researchers attempting to reproduce and build upon this work.
3. While synergy is claimed, the mechanism of mutual benefit (how disentangled features improve RL stability and vice versa) is discussed qualitatively but not empirically isolated. Table 3 does not show results with only RL applied (without IDM). It's better to provide these results to support the qualitatively analysis.

4. Minor issue: Some notations (e.g., R+, R−, CAS) could be made clearer.

**Questions:**

See Weakness.

---

> ### Author Response · Authors · 2025-11-21
> **Response to Reviewer WrbM**
>
> Thanks for acknowledging our contribution to L2I in terms of creativity and performance. Our point-to-point responses are as follow.
>
> **Q1**. Using GDINO as a fixed reward model introduces bias and limits generalization. The paper does not examine robustness when GDINO fails to detect objects or misinterprets captions.
>
> **A1**. To check our method against RL's false reward, we conduct experiments when setting reward to its negative, *i.e.*, deliberately assigning false reward for mimicking the false signal from GDINO to check I-DRUID's robustness. The results of ISR on COCO-MIG are as follow.
>
> **Tab. ** Robustness to False Reward on COCO-MIG.
>
> | ISR | L2 | L3 | L4 | L5 | L6 |
> | :---: | :---: | :---: | :---: | :---: | :---: |
> | w/o False Reward | 76.87 | 69.16 | 62.96 | 52.37 | 52.39 |
> | w/ False Reward | 67.18 | 66.25 | 57.34 | 51.20 | 51.92 |
>
> From the result, we conclude that our RL part suffers from slight performance degradation when wrong rewards are given. This is due to two reasons:
>
> (1) **Joint optimization with paired data and KL terms**. This is a commonly used strategy [a-c] to overcome training instability during RL. By using KL term and keeping original pair-wise training loss, our method mitigates the negative effects brought by RL. Moreover, we only conduct RL at the last 10 epochs of training, where our model already has strong capability for L2I generation. During the trajectory sampling, the generated images will not introduce samples with completely false layout, alleviating the negative effects brought by false reward.
>
> (2) **Feature Disentanglement**. As demonstrated in Tab.3 of our paper, after using IDM for RL, the final L2I accuracies are improved. Therefore, our IDM provides better representation for subsequent RL.
>
> Although our method achieves good stability, to further alleviate the negative effects brought by false reward, one can combine multiple reward functions (combining multiple object detectors, *etc.*) to improve reward robustness, or carefully adjusting KL weights to strike a balance between aggressive exploration and stable training. To sum up, our method brings consistent improvement on L2I (higher ISR, mIoU, and spatial scores in Tab.1 and lower FID in Tab. 2), while negative effects brought by reward also have solutions. We have added these discussion in our revision, at Appendix--Sec. G: "Robustness to Unreliable Reward".
>
> **Q2**. The framework is too complex and makes this work hard to follow.
>
> **A2**. The overall training process of our method has been provided in the Appendix--Sec. A: "Overall Training Pipeline". To improve the reproducibility, we further provide the pytorch style pseudo code of the implementation at Appendix--Sec. H: "Pseudo Code of Our I-DRUID". Moreover, code will be available if the paper is accepted.
>
> **Q3**. Table 3 does not show results with only RL applied (without IDM). It's better to provide these results to support the qualitatively analysis.
>
> **A3**. Thanks. We report the results of RL-only variant in our revised manuscript. Specifically, when solely using RL for L2I optimization, the results are as follow. COCO-MIG Avg ISR: 60.23; LayoutSAM-eval: Spatial=91.47; Color=72.45; Texture=74.79; Shape=72.93. The results further demonstrate the efficacy of RL. Please find them in our revision, at Tab.3.
>
> **Q4**. Minor issue: Some notations (e.g., R+, R-, CAS) could be made clearer.
>
> **A4**. We have made these notations clearer in the revision by adding more explanation at the caption of Fig.2. Please refer to our revised manuscript for details.
>
> [a] Fine-Tuning Language Models from Human Preferences. In Arxiv'19.
>
> [b] DPOK: Reinforcement Learning for Fine-tuning Text-to-Image Diffusion Models. In NIPS'23.
>
> [c] Training Diffusion Models with Reinforcement Learning. In ICLR'24.

---

### Author Response · Authors · 2025-11-21
**Rebuttal Summary for Submission 684**

## Dear Area Chair,

We sincerely appreciate your consideration of our submission, and we thank to all reviewers for their insightful and valuable feedback. Below we summarize the contributions and response to the reviewers' concerns.


### **I. Summary of Contributions**

**(1) Novel Framework for L2I with SoTA Performance**. We propose a novel framework I-DRUID for layout-to-image generation (L2I), seeking semantic-related features for accurate instance-level control while improving model's generalization with unpaired data. The framework achieves SoTA performance while reduces annotation costs, which is acknowledged by most reviewers (Reviewer h5RF, WrbM, and uTfS).


**(2) Thorough Evaluation and Ablation Study**. We conduct extensive experiments on two large-scale L2I benchmarks, while providing detailed ablation studies to show the effectiveness of our method (acknowledged by Reviewer h5RF, WrbM, and uTfS).


### **II. Summary of Reviewer's Non-overlapped Comments and Our Key Updates**

All suggestions have been seriously considered, and we have polished our manuscript based on reviewers' comments. We sumarize each reviewer's non-overlapped concerns as follow.

#### **Reviewer WrbM**

(1) Robustness to Unreliable Reward. We confirm our method's minor sensitivity to false reward and provide several practical solutions to further alleviate it (Appendix -- Sec. G).

(2) Reproducibility. We add pseudo code of our core implementation at appendix (Appendix -- Sec. H).

(3) More Ablation. We report the requested results to Tab. 3.

#### **Reviewer h5RF**

(1) Runtime and Generic Image Quality Metrics. We report the runtime (Sec. 4.1 Implementation Details) and image quality metrics (Tab.2).

(2) More explanation on SD3 and SD1.5 Results. We add more explanation to the L2I results under SD3 and SD1.5 scenario (Appendix--Sec. C).

#### **Reviewer 6mc3**

(1) Evidence of Alleviating Attribute Leakage. We add more visualization to demonstrate the alleviation of attribute leakage (Sec. 4.4--"Ablated Visual Comparison").

(2) Justification of Reward Design. We add more quantitative and qualitative experiments (at Appendix--Sec. D.) to show the rationale behind our reward design.

(3) Novelty Clarification. We explain our method's novelty in terms of motivation and methodology. *In terms of motivation*, Our method is motivated by the need to enhance generalization with unpaired, prompt-only data, which is under-explored in concurrent works and acknowledged by Reviewer uTfS and h5RF. *In terms of technology*, Our method proposes to conduct RL on semantic-related policy space, which is proven to be a better policy space than standard feature space for L2I. Although RL itself is based on previous works, conducting RL on semantic-related feature space while obtaining such features with our IDM is original. By clarifying the necessity of our technology and the innovative motivation behind utilizing unpaired data, we hope to demonstrate the non-trivial design and novelty of our method. We sincerely appreciate reviewer h5RF, WrbM, and uTfS in acknowledging our novelty and contributions. We would also like to kindly ask AC to realize the non-trivial design of our framework.

(4) Clarification of Notations. We add more explanations to our notations (Sec.3.2). We believe the revised manuscript has addressed Reviewer 6mc3's concern regarding our loss design.


#### **Reviewer uTfS**

(1) Lack of Comparable SoTA Methods. We add comparison with "RichContext", a latest L2I method, at Tab.1.

(2) Clarification of Duplicate Risk. We have addressed the concern by mentioning our experimental setting, which is acknowledged by the reviewer.

The revised part are highlighted with orange in the revision. We believe our revised manuscript has thoroughly addressed all reviewer concerns. We hope the Area Chair finds our responses satisfactory.

Sincerely,

The Authors of Submission 684

---

### Meta-Review · Area_Chair_yDWP · 2026-01-05

**Summary:**

This paper proposes I-DRUID, a layout-to-image (L2I) generation framework that targets two core challenges: attribute leakage across instances and limited generalization to novel scenes due to the scarcity of paired image–text data. The method consists of two main components. First, an instance disentanglement module and constraint are introduced to separate semantic-related features from spurious parts, aiming to improve instance-level control and reduce attribute leakage. Second, reinforcement learning with PPO is applied on unpaired, prompt-only data, using AI feedback (e.g., Grounding-DINO–based rewards) to adapt the L2I model without requiring additional paired data collection. During the rebuttal, the authors added missing ablations (including an RL-only variant), robustness analysis to unreliable rewards, clearer notation and pseudocode, runtime reporting, and expanded comparisons, which significantly strengthen the empirical evidence and improve reproducibility.

**Reviewer Concerns:**

Concerns addressed in the rebuttal.

1.	Robustness to unreliable rewards. The authors added experiments with incorrect rewards and showed only moderate performance degradation, along with stabilization strategies such as joint supervised training, KL regularization, and late-stage RL.

2.	Missing ablation analysis. Additional ablations, including an RL-only baseline, were provided to better disentangle the effects of IDM, RL, and their combination.

3.	Reward design. Both quantitative and qualitative comparisons of different reward formulations were included, supporting the default IoU-plus-confidence choice.

4.	Reproducibility and clarity. Pseudocode, more explicit training descriptions, and notation clarifications were added to improve readability and reproducibility.

5.	Efficiency and evaluation. Runtime statistics and additional image-quality metrics beyond geometric accuracy were reported.

6.	Comparable baselines. Comparisons with a recent L2I method were added, and fairness issues across different backbones were clarified.

Concerns still outstanding.

1.	Novelty and framing of the RL component. The RL contribution may still be viewed as an incremental adaptation of existing diffusion-RL techniques, despite the authors’ arguments on semantically related policy spaces.

2.	Reward bias and evaluation validity. Using a detection-based reward may bias optimization toward spatial grounding while under-emphasizing realism or text fidelity, and concerns about reward generalization may remain.

3.	Heuristic nature of the disentanglement constraint. The CAS-based formulation remains heuristic mainly, and its necessity or optimality has not been fully established beyond empirical results.

4.	Method complexity. Despite added clarifications, the overall framework remains complex and may pose challenges for reproduction.

**Reviewer Scores:**

Reviewer uTfS: likely stays at 6, as the reviewer explicitly indicated an inclination toward acceptance after the revisions.

Reviewer h5RF: likely stays at 6, since the main concerns regarding runtime, evaluation metrics, reward stability, and SD-1.5 versus SD-3 behavior were largely addressed.

Reviewer WrbM: may move from 4 to 6 due to the added robustness analysis, RL-only ablation, and improved reproducibility, but could also remain at 4 if reliance on the reward model remains a decisive concern.

Reviewer 6mc3: likely remains in the 4–6 range, as concerns about the novelty of the RL component and the rationale of the disentanglement constraint appear only partially alleviated.

---

### Decision · Program_Chairs · 2026-01-26

Accept (Poster)